# A vertical silicon-graphene-germanium transistor

Chi Liu[1,3], Wei Ma [1,2,3], Maolin Chen[1,2], Wencai Ren [1,2] & Dongming Sun [1,2]*

Graphene-base transistors have been proposed for high-frequency applications because of the negligible base transit time induced by the atomic thickness of graphene. However, generally used tunnel emitters suffer from high emitter potential-barrier-height which limits the transistor performance towards terahertz operation. To overcome this issue, a graphene-base heterojunction transistor has been proposed theoretically where the graphene base is sandwiched by silicon layers. Here we demonstrate a vertical silicon-graphene-germanium transistor where a Schottky emitter constructed by single-crystal silicon and single-layer graphene is achieved. Such Schottky emitter shows a current of $692\,A\,cm^{-2}$ and a capacitance of $41\,nF\,cm^{-2}$, and thus the alpha cut-off frequency of the transistor is expected to increase from about $1\,MHz$ by using the previous tunnel emitters to above $1\,GHz$ by using the current Schottky emitter. With further engineering, the semiconductor-graphene-semiconductor transistor is expected to be one of the most promising devices for ultra-high frequency operation.

[1] Shenyang National Laboratory for Materials Science, Institute of Metal Research, Chinese Academy of Sciences, 72 Wenhua Road, Shenyang 110016, China. [2] School of Materials Science and Engineering, University of Science and Technology of China, 72 Wenhua Road, Shenyang 110016, China. [3] These authors contributed equally: Chi Liu, Wei Ma. *email: dmsun@imr.ac.cn

In 1947, the first transistor, named a bipolar junction transistor (BJT), was invented in the Bell Laboratory and has since led to the new age of information technology. In an n-p-n BJT, a p-type base semiconductor is sandwiched by n-type emitter and collector semiconductors, forming an emitter junction between the emitter and the base, and a collector junction between the base and the collector. During operation, electrons are emitted from the emitter, diffuse through the base, and eventually are collected by the collector[1].

For a BJT, a main figure of merit is the alpha cutoff frequency $f_\alpha$, which is used to represent the upper frequency limit when a BJT is biased in the common base mode. $f_\alpha$ is inversely proportional to the delay time, which includes the emitter charging time $\tau_e$, the base transit time $\tau_b$, and the collector delay time $\tau_c$[2–5]. In the past decades, there has been a persistent demand for higher frequency operation for a BJT, leading to the inventions of new devices, such as heterojunction bipolar transistors and hot electron transistors. The heterojunction bipolar transistors have achieved great development toward the terahertz (THz) operation[6–9], however, their $f_\alpha$ is ultimately limited by $\tau_b$. For the hot electron transistors, when an electron is emitted into the base, the energy difference between the electron and the Fermi energy level (or the bottom of conduction band) of the base is transformed into electron kinetic energy. This makes the electron hot with a high speed and leads to a small $\tau_b$[10–14]. However, the demand of a thin base without pinholes and with a low base resistance usually causes difficulties in material selection and fabrication. Recently, to reduce $\tau_b$, graphene has been proposed as a base material to form graphene-base transistors[15]. Because of the atomic thickness, the graphene base is almost transparent to electron transport leading to a negligible $\tau_b$. At the same time, the remarkably high carrier mobility of graphene will benefit the base resistance compared with a thin bulk material[16,17].

So far, the proposed graphene-base transistors generally use a tunnel emitter, which emits an electron through an insulator[18–23]. As predicted by simulations, in order to realize THz operation, the emitter potential barrier which is between the emitter metal and the emitter-to-base tunneling layer should be <0.4 eV, which remains an engineering issue[15,24,25]. To solve this problem, pioneering theoretical study on graphene-base heterojunction transistors has been done with a device structure of silicon–graphene–silicon[25,26]. Theoretically, THz operation can be realized when the collector current is >$10^6$ A cm$^{-2}$.

Here, we demonstrate a vertical silicon–graphene–germanium (Si–Gr–Ge) transistor. The Si–Gr Schottky emitter carries a current of 692 A cm$^{-2}$ at 5 V, and has a capacitance of 41 nF cm$^{-2}$. As a result, $f_\alpha$ is expected to increase from ~1 MHz by using the previous tunnel emitters to >1 GHz by using the current Schottky emitter. With further engineering, the vertical semiconductor–graphene–semiconductor transistor is expected as one of the most promising devices for ultra-high-frequency operation in future 3D monolithic integration, because it combines the merits of the atomic thickness and high carrier mobility of graphene, and the high feasibility of a Schottky emitter.

## Results

### Device design and fabrication

Figure 1a shows a schematic of Si–Gr–Ge transistor fabrication using semiconductor membrane and graphene transfer[27–33]. Three materials are directly stacked, including an n-type top single-crystal Si membrane, a middle single-layer graphene, and an n-type bottom Ge substrate (see the Methods section). An optical image of a fabricated Si–Gr–Ge transistor is shown in Fig. 1b, where graphene is patterned to isolate each device. The Si membrane with a deposited Au electrode on it closely contacts the graphene as shown in the scanning

electron microscope (SEM) image in Fig. 1c. Figure 1d shows a cross-section of a transistor, where Si, Gr, and Ge are stacked in the window of a dielectric layer. As shown in Fig. 1e, emitter and collector Schottky junctions are formed due to the work function differences between the graphene and the semiconductors. When the device is turned on with a positive voltage $V_{be}$ applied to the emitter junction (when graphene is grounded, $V_e < 0$), electrons are emitted from the emitter, go through the emitter junction with a barrier height $q\phi_1$, the graphene base, then the collector junction with a lower barrier height $q\phi_2$, and eventually are collected by the collector.

### Schottky emitter of the Si–Gr–Ge transistor

Figure 2a shows a typical current–voltage (I–V) characteristic of the top Si–Gr emitter, achieving an on-to-off current ratio of $1.8 \times 10^6$ at ±5 V. Obvious rectifying behavior indicates the formation of a Schottky barrier. Note that the Ohmic contact between the Au electrode and top Si is formed (Supplementary Figs. 1, 2). The current shows an obvious temperature dependence (Supplementary Fig. 3), which is a feature of the thermionic emission of a Schottky junction, and different from the temperature-independent feature of the current of a tunnel junction. In Fig. 2b, an Arrhenius plot of current vs temperature at a voltage of −0.1 V reveals a Schottky barrier height of 0.64 eV at room temperature. A Schottky barrier height of 0.68 eV at a voltage of 0 V is achieved by an extrapolation method (Supplementary Fig. 3). This is the first Si-membrane-on-Gr junction, in which the ideality factor and the Schottky barrier height are consistent with those of Gr-on-Si junctions previously reported in literature[34–42] (Supplementary Table 1).

In a transistor, the emitter charging time $\tau_e$ is proportional to the emitter capacitance $C_e$, and inversely proportional to the emitter conductance $g_e$. A high on-current of the Schottky emitter is necessary for high-frequency applications, since it leads to a large emitter conductance $g_e$ and therefore may contribute to a small emitter charging time $\tau_e$. In Fig. 2c, the on-current of our Si–Gr Schottky emitter is compared with those of previously reported tunnel emitters of graphene-base transistors. Because of the strong exponential dependence of the emitter current on voltage, the Si–Gr emitter shows an on-current of 692 A cm$^{-2}$ at −5 V, leading to a $g_e$ of 347 S cm$^{-2}$. On the other hand, for a tunnel emitter, the generally used several-nanometer-thick insulator leads to a large emitter capacitance $C_e$ which increases $\tau_e$ (Supplementary Table 2). However, a small capacitance can be achieved by a Schottky junction with a lightly doped semiconductor. The emitter capacitance of our Si–Gr Schottky emitter[43–47] is at most 41 nF cm$^{-2}$, which is at least one-order-of-magnitude lower than those of its tunnel emitter counterparts (Supplementary Fig. 4).

For the Si–Gr emitter, $\tau_e = C_e/g_e$ is ~118 ps at −5 V leading to an $f_\alpha$ of 1.2 GHz, which indicates that the Schottky emitter can take the cutoff frequency of a graphene-base transistor into the GHz range. Here, $f_\alpha$ is estimated by $f_\alpha = 1/[2\pi(\tau_e + \tau_b + \tau_c)]$, where $\tau_e$ is a dominating factor (Supplementary Fig. 5). Figure 2d shows a comparison of $f_\alpha$ of graphene-base transistors with different emitters, where a clear frequency gap is observed between the ones using tunnel emitters (up to 1 MHz) and the one using the Schottky emitter (up to 1.2 GHz).

### Electrical characteristics of the Si–Gr–Ge transistor

The transistor is biased in the common base mode, where graphene is connected to ground (Supplementary Fig. 6). $I_e$ is emitted from the Si–Gr emitter, and part of it ($I_c'$) is collected at the Ge collector which is called the effective collector current in an amplifier. A leakage current $I_{leak}$ of the collector junction also

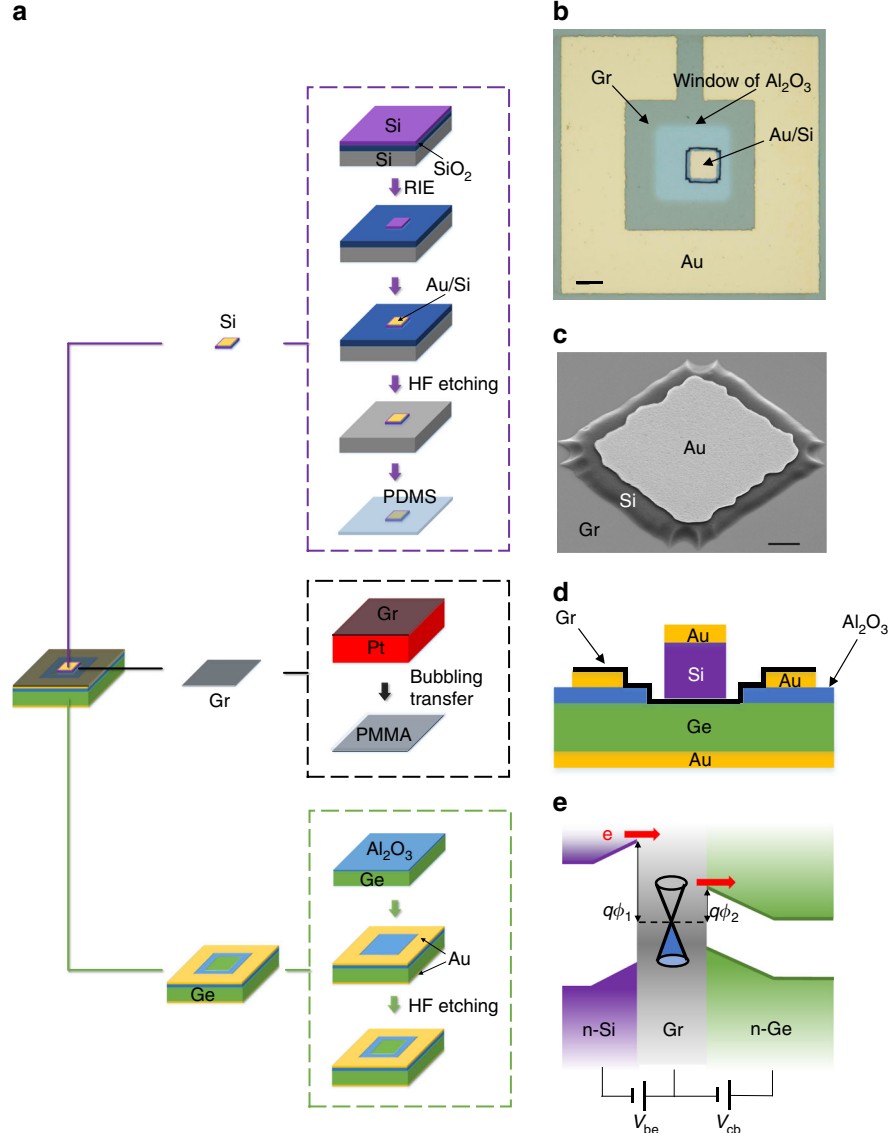

**Fig. 1** Device design and fabrication. **a** A Si–Gr–Ge transistor is built by directly stacking a Si membrane, single-layer graphene and a Ge substrate. **b** Optical image of a Si–Gr–Ge transistor (scale bar: 20 μm). **c** SEM image of a Si membrane on graphene (scale bar: 4 μm). **d** Illustration of the cross-section of the transistor. **e** Illustration of the basic operating principle of the transistor

contributes to the collector current $I_c$. The common base current gain is calculated as $\alpha = I_{c'}/I_e = (I_c - I_{leak})/I_e$ and $I_{leak} = I_c$ ($V_e = 0$). Figure 3a shows the I–V characteristics of the Si–Gr and Gr-n-Ge junctions in a Si–Gr–Ge transistor with a lightly doped n-Ge collector (resistivity: ~1 Ω cm). The Schottky barrier height of Gr–Ge junction is estimated to be 0.22 eV (Supplementary Fig. 7). As shown in the input and transfer characteristics, when $I_e$ increases, $I_c$ increases accordingly, indicating a successful collection of part of the emitted electrons (Fig. 3b). This collection is more obviously seen from the increasing $I_{c'}$ with $I_e$ (Fig. 3c) and the current gain is ~1% (inset of Fig. 3c). This collection can also be seen from the output characteristics, where $I_c$ increases as the input current $I_e$ increases (Fig. 3d).

To increase the current gain, a heavily-doped $n^+$-Ge collector (resistivity: ~0.1 Ω cm) is used. This results in a much-increased electrical field at the collector junction (Supplementary Fig. 8). At the collector junction interface, around the top of the barrier, the tunneling distance of an electron decreases dramatically. As a result, electrons can tunnel through the barrier even if they cannot cross it, which increases the current gain. Figure 3e shows

the I–V characteristics of the Si–Gr and Gr-n$^+$-Ge junctions, where the leakage current of the Ge junction increases with reverse bias because of the enhanced tunneling effect. Figure 3f shows the input and transfer characteristics. $I_c$ increases very closely to $I_e$ when a large $V_c$ is applied. This is more obvious in Fig. 3g where $I_{c'}$ is shown. As shown in the inset of Fig. 3g, when $V_c > 3$ V, the current gain increases toward 100% with $V_e$. Figure 3h shows the output characteristics with different input current $I_e$. The output characteristics with different voltage $V_e$ is also shown in Supplementary Fig. 9, from which and Fig. 3f, a region with the power gain estimated >1 can be observed.

A trade-off exists between the high output impedance using a lightly doped collector and the large current gain using a heavily doped one. A fundamental solution to obtain both a high-output impedance and a large current gain is to increase the quality of the interfaces to reduce the interface scattering[48]. On the other hand, to reduce the collector junction leakage, material engineering is needed (Supplementary Discussion 1).

For the transistor with an $n^+$-Ge collector, graphene-base currents are shown to support the above analysis (Supplementary

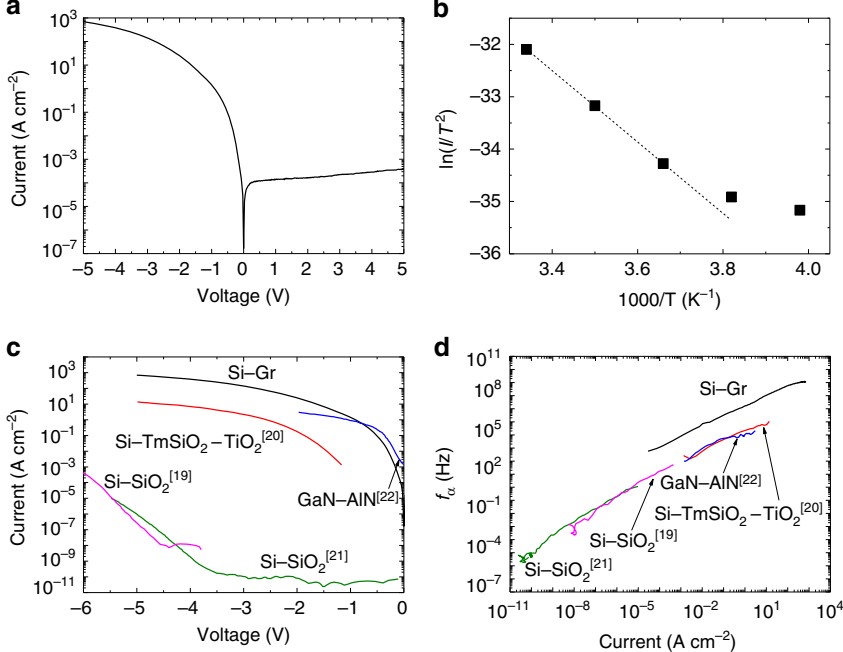

**Fig. 2** Schottky emitter of the Si–Gr-Ge transistor. **a** A typical I–V characteristic of the top Si–Gr emitter junction at room temperature showing an obvious rectifying behavior. **b** Temperature-dependent characteristics of the current. An Arrhenius plot at a voltage of $-0.1$ V gives a Schottky barrier height of 0.64 eV at room temperature. **c** Comparison of the on-currents of graphene-base transistors with different emitters. The Si–Gr Schottky emitter shows a current of 692 A cm$^{-2}$ at $-5$ V. **d** Comparison of $f_\alpha$ of graphene-base transistors with different emitters. The one with the Si–Gr Schottky emitter shows the best cutoff frequency of 1.2 GHz

Fig. 10). For the transistor with an n-Ge collector, the small $V_c$ dependence of $I_e$ helps eliminate the concern that the emitter and collector may be in contact (Fig. 3b). On the other hand, $I_e$ is dependent on $V_c$ for the transistor with an n$^+$-Ge collector (Fig. 3f). The transfer characteristics of the transistor when the emitter and collector are exchanged indicate that the emitter and the collector are not in direct contact (Supplementary Fig. 11).

## Discussion

Figure 4 shows the energy band diagram considering the quantum capacitance of graphene[49] to further investigate the operating mechanisms of the transistor. As shown in Fig. 4a, when no bias is applied, the Fermi energy level (FEL) remains flat in the transistor. The FEL is assumed to go through the Dirac point, which does not affect the analysis. For clear explanation, the effects of $V_{be}$ and $V_{cb}$ are considered separately, but can be easily combined.

As shown in Fig. 4b, when a forward bias is applied to the emitter junction, the conduction band and the FEL of Si move upward. Because of the quantum capacitance effect of graphene, its FEL moves down from the Dirac point. The FEL remains flat in the collector junction since no bias is applied. As a result, the conduction band energy and the FEL of the collector Ge move down with the FEL of graphene. The distance between the FELs of Si and Ge is $qV_{be}$. Finally, the injected electrons from Si into Gr lift the FEL in Gr, resulting in a FEL difference between Gr and Ge.

Similarly, as shown in Fig. 4c, when a reverse bias is applied to the collector junction, a FEL difference is generated between Gr and Ge, and part of the bias is applied to Gr because of the quantum capacitance effect. The conduction band energy and the FEL of the emitter Si move up with the FEL of graphene, and the FEL difference between Si and Ge becomes $qV_{cb}$. For the n$^+$-Ge collector, the large reverse tunneling current pumps out electrons,

lowering the FEL of Gr. This induces a FEL difference between Si and Gr.

Using the above energy band diagrams, several phenomena can be understood, including the bias-dependent current gain and $V_c$-dependent $I_e$ of the transistor with an n$^+$-Ge collector (Supplementary Discussion 2). When a large bias is applied to the emitter junction, most of the bias will be applied to the series resistance of the junction (Supplementary Fig. 12). Based on the quantum capacitance effect of graphene, a reverse working mode (Ge is used as input, and Si is used as output but with $V_c > 0$, $V_e < 0$) of the transistor is possible (Supplementary Figs. 13, 14).

For the Si–Gr emitter, the conductance can be further increased by a reduction of the series resistance and interface engineering. Based on the experimental results, an ideal case with an ideal interface, ignoring the series resistance, is considered for a transistor with a heavily doped n$^+$-Si–Gr emitter and a thin collector, and THz operation is expected by using this Schottky emitter with an emitter current of $\sim 2.1 \times 10^6$ A cm$^{-2}$, which is consistent with the theoretical predictions[25] (Supplementary Fig. 15).

In conclusion, a Si–Gr–Ge transistor has been demonstrated. The Schottky emitter provides an emitter charging time of ~118 ps with a current of 692 A cm$^{-2}$ and a capacitance of 41 nF cm$^{-2}$, which is expected to increase the alpha cutoff frequency from the previous ~1 MHz by using tunnel emitters to >1 GHz by using the Schottky emitter in a graphene-base transistor. With further engineering, the vertical semiconductor–graphene–semiconductor transistor is promising for high-speed applications in future 3D monolithic integration because of the advantages of the atomic thickness and high carrier mobility of graphene, and the high feasibility of a Schottky emitter.

## Methods

**Fabrication of the Si membrane**. A silicon-on-insulator (SOI) substrate with a 2-μm-thick top single-crystal Si layer (resistivity: ~4.5 Ω cm), and a 1-μm-thick

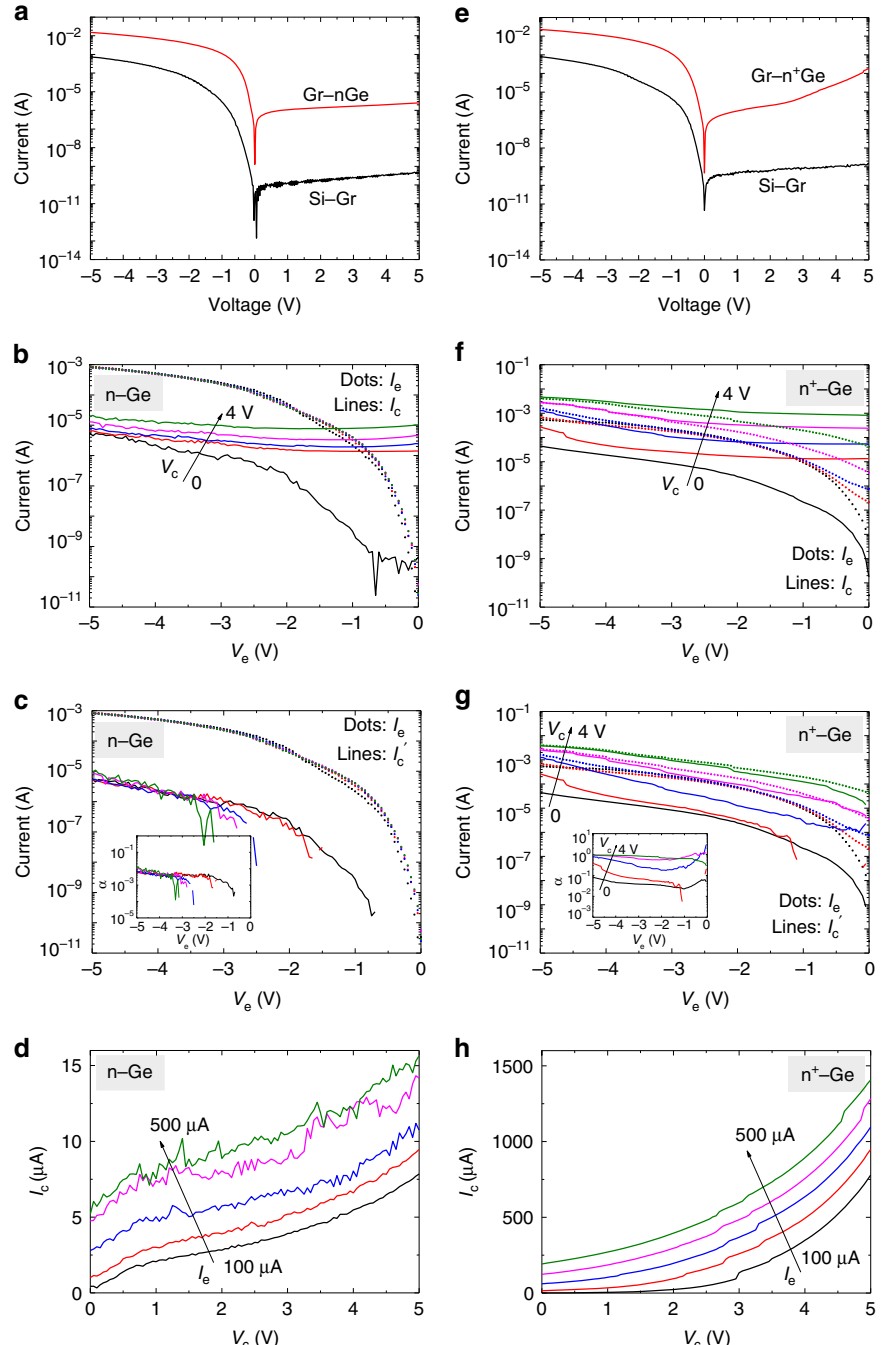

**Fig. 3** Electrical characteristics of the Si-Gr-Ge transistors in the common base mode. The figures in left column (**a**–**d**) are for the transistor using a lightly doped n-Ge collector, and those in the right column (**e**–**h**) are for a heavily-doped n+-Ge collector. **a** The I–V characteristics of the Si-Gr and Gr-n-Ge junctions. **b** Input ($I_e - V_e$) and transfer ($I_c - V_e$) characteristics where $V_c$ changes from 0 to 4 V. **c** Transfer ($I_c\prime - V_e$) characteristics after eliminating the influence of the collector junction leakage. Inset: common base current gain. **d** Output ($I_c$-$V_c$) characteristics. $I_e$ changes from 100 to 500 μA. **e**–**h** Corresponding electrical characteristics of the transistor using a heavily doped n+-Ge collector

buried $SiO_2$ layer was used as a membrane provider. For photolithography, photoresist s-1813 (spin-coated at 3000 r.p.m. for 30 s, baked at 120 °C for 2 min) and LOR7A (spin-coated at 3000 r.p.m. for 50 s, baked at 190 °C for 5 min) were used in sequence. The top Si layer was patterned into ~26 × 26 μm² squares by reactive ion etching (RIE) with the following parameters: $CF_4$: $O_2$ = 25 sccm: 5 sccm, 13.3 Pa, RF power = 100 W, for 10 min. The height of the Si membrane was more than 800 nm as measured by an atomic force microscopy (AFM) (Supplementary Fig. 16). A 50-nm-thick Au electrode was deposited on the top surface of the Si membrane, and then the device was annealed in Ar at 350 °C for 30 min. Ohmic contact was obtained between the Au electrode and the Si membrane (Supplementary Figs. 1, 2). Concentrated HF acid (40 wt%) was used to etch away the $SiO_2$ (15 min), leaving the patterned top Si membranes onto the bottom Si substrate.

**Transfer of the Si membrane**. For transfer of the Si membrane, PDMS was used as a medium to transfer the top Si membrane onto the target substrate. The PDMS was first prepared by stirring a mixed solution of PDMS and its curing agent (10: 1 in weight) and baking at 65 °C for 6 h. To retrieve the Si membrane[27], the PDMS was placed on the bottom Si substrate with the Si membrane, and peeled away quickly at a rate >10 cm s⁻¹. To release the Si membrane[27], the PDMS with the Si membrane was placed on the target substrate, and peeled away slowly at a rate <1 cm s⁻¹.

**Fabrication of the single-layer graphene film**. A 20 × 10 mm² piece of Pt foil (250-μm-thick, Alfa, 99.99 wt% metal) was rinsed with DI water, acetone, and ethanol in sequence for 1 h each, loaded into a quartz holder inside the fused silica

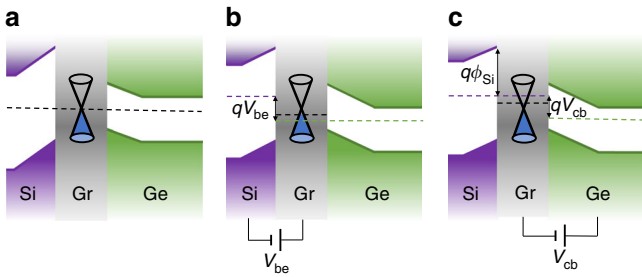

**Fig. 4** Energy band diagrams of the Si–Gr–Ge transistor. **a** Energy bands when no bias is applied. **b** Energy bands when a forward bias $V_{be} > 0$ is applied to the emitter junction. **c** Energy bands when a reverse bias $V_{cb} > 0$ is applied to the collector junction

tube (inner diameter: 22 mm) of a tube furnace (Lindberg Blue M, Thermo Scientific), and then annealed at 1000 °C for 10 min to remove residual carbon or organic substances under a hydrogen atmosphere. Growth was then initiated and maintained for 40 min under a mixture of $CH_4$ (4.5 sccm) and $H_2$ (500 sccm) at 1000 °C. Finally, the Pt foil was quickly pulled out of the high-temperature zone, and the $CH_4$ flow was turned off when the furnace temperature was below 800 °C[29–31].

**Transfer of the single-layer graphene film.** For bubbling transfer of the graphene film, a Pt substrate with the grown graphene was spin-coated with PMMA (950 kDa molecular weight, Sigma, 4 wt% in ethyl lactate) at 2000 r.p.m. for 60 s, and then cured at 180 °C for 15 min. A constant current of 0.2 A was applied to separate the PMMA/graphene layer from the Pt foil in a 1 M NaOH aqueous solution. The PMMA/graphene films were then collected on the target substrates, and finally acetone was used to remove the PMMA at 50 °C[29–31].

**Patterning of the Ge substrate.** A 30-nm-thick $Al_2O_3$ insulator layer was deposited on top of an n-type Ge substrate (resistivity: ~1 Ω cm for a lightly doped sample, ~0.1 Ω cm for a heavily doped one) by atomic layer deposition (ALD) at 150 °C (precursors: trimethylaluminum (TMA) and water). Au electrode was formed on the top of $Al_2O_3$. The bottom of the Ge substrate was scratched, and Au metallization was performed to form an Ohmic contact. The $Al_2O_3$ layer was patterned by photolithography and etching with dilute HF (5 wt%) for 3 min, leaving an ~60 × 60 μm² window to the Ge substrate.

**Fabrication of the Si–Gr–Ge transistor.** Single-layer graphene and Si membranes were fabricated and transferred to the Ge substrate one after the other. The graphene was patterned to ensure isolation of each device by photolithography and $O_2$ plasma etching (200 W, 180 sccm, 2 min).

**Characterization.** The fabricated devices were characterized using an optical microscope (Nikon LV100ND), an SEM (FEI XL30 SFEG using an accelerating voltage of 10 kV), and an AFM (Bruker Dimension Icon AFM). The electrical performance of the transistors was measured using a semiconductor analyzer (Agilent B1500A with a capacitance measurement unit B1500A-A20) and a probe station (Cascade Microtech Inc. 150-PK-PROMOTION) under ambient conditions, and a vacuum probe station (Lake Shore TTPX/TSM1D1001) under low-temperature conditions. About 10–30 working transistors are successfully fabricated and characterized for each wafer with a yield of 10–30% (Supplementary Fig. 17). To enhance the yield, larger and cleaner graphene without damage and SOI wafers with more uniform top Si and oxide layers in thickness should be used.

## Data availability

The data that support the findings within this study are available from the corresponding author upon reasonable request.

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

## Acknowledgements

This work is supported by National Natural Science Foundation of China (Nos 61704175, 51625203, 51532008, 51521091, 51272257, 51572264, 51502304, 61422406, 61574143, 51325205, 51290273, and 51521091), Institute of Metal Research, Chinese Academy of Sciences (project Young Merit Scholars and No. 2017-PY04), the Chinese Academy of Sciences (Grant KGZD-EW-T06, ZDBS-LY-JSC027), the Strategic Priority Research Program of Chinese Academy of Sciences (Grant No. XDB30000000), the Thousand Talent Program for Young Outstanding Scientists, and the National Key Research and Development Program of China (2016YFB0401104, 2016YFA0200101). The authors wish to thank Hui-Ming Cheng, Peter Thrower, Yi-Peng Wang, Hua-Hua Li, Hai-Jun Lou, Takeaki Yajima, Tomonori Nishimura, Akira Toriumi, and Lin Zhu for valuable discussions.

## Author contributions

C.L. and D.S. conceived the project. C.L. and W.M. contributed equally to this work. C.L. designed and performed the experiments, as well as electrical measurements and data analysis. W.M. carried out graphene growth and transfer supervised by W.R. M.C. performed SEM and AFM measurement. C.L. and D.S. wrote the paper. All authors discussed the results and commented on the paper.

## Competing interests

The authors declare no competing interests.
