## [Peer Review File · Nature Communications]

Reviewers' comments:

Reviewer #1 (Remarks to the Author):

The authors fabricated a graphene-base transistor with Si emitter and Ge collector. The main claims of the paper are record current density and potential for THz operation. However, I find that the claims are not well supported by the experimental data and are incorrect in some cases. The manuscript is not suitable for publication in the current format. Here are my comments,

1. The authors experimentally demonstrated the graphene-base heterojunction transistor proposed by Di Leece et al. (IEEE TRANSACTIONS ON ELECTRON DEVICES, VOL. 60, NO. 12, DECEMBER 2013). It would be useful to discuss the performance gap between the theoretical results proposed by Di Leece et al. and the experimental results presented in this paper.
2. Graphene/Si schottky diode is a widely studied structure (Di Bartolomeo , Physics Reports Vol. 606, Jan 2016, Pp 1-58). Can the authors benchmark their Schottky diode against the literature?
3. I'm not convinced that the contact between probe and Si is ohmic. The resistivity of the Si is too high (6 ohm.cm) to make ohmic contact just by landing probes on it. The I-V presented in the supplementary is linear but current level is too low compared to graphene if I assume same area. The authors need to fabricate proper ohmic contact to study the device operation. Otherwise, the device operation is limited by the contact resistance between probe and Si.
4. Authors claim that they have achieved record current density. However, from Fig. 2(c) it is obvious that current work has similar or lower current when compared with ref 17 for the same applied bias and it can achieve higher current density only when the applied bias is larger. Moreover, Guo et al. (IEEE Electron Device Letters, Vol.39 , No. 4 , pp. 634-637, April 2018) have demonstrated similar current density (~ 233 A/cm²) with much smaller applied bias across tunnel emitter. Could the authors please comment on that?
5. I'm confused by the primed (V_c' , V_e') and regular notations (V_c and V_e). Could the authors please clarify the difference? It is hard to understand the device operation without clarification.
6. How does the base-collector cold electron leakage affect the gain of the device? Did the authors consider that while calculating the gain in Fig 3 c? Is the diode characteristics presented in Supp. Fig. 3 b is from the same device of Fig. 3b? Please show the base current in Fig 3b. I_e and I_c should be swapped in Fig 3d.
7. The authors mentioned that their device can potentially overcome the frequency limitation of BJT and predicted THz operation for optimized device geometry without considering any non-idealities. InP HBT technologies have already reached $f_t > 0.5$ THz and $f_{max} > 1$ THz (M. Urteaga et al. 69th Device Research Conference , 2011). Could the authors please compare their projection with the InP HBT experimental results before claiming improvement over the state-of-the-art?

8. The authors also mentioned that it would require 1 μ m thick emitter to achieve THz operation. Would it increase the emitter transit time?

Reviewer #2 (Remarks to the Author):

The paper reports on a successful fabrication of vertical silicon-graphene-germanium transistors with performance that notably surpasses other results published so far. Based on the measured data, the authors predict for such devices high-frequency operation with cut-off frequency of 1 THz at 1 kA/cm², that is, at current levels comparable to those achieved by the devices they made. This makes the concept of a graphene base transistor (GBT), i.e., of a transistor in which the input-output current flows in the direction normal to the graphene sheet, a realistic alternative to the usually studied concept of a graphene field-effect transistor, i.e., of a transistor in which the input-output current flows in the plane of the graphene sheet. This achievement is therefore of sufficient novelty and of sufficient general interest to justify the publication of the report in Nature Communications.

Nevertheless, my opinion is that the distribution of accents in the main message as well as in the write-up should be optimized. The paper would also benefit from some improvements in the presentation of the acquired data and in the discussion. There are several points to be addressed, so I recommend publication of the paper after a major revision.

1. The importance of the hot-electron character of the device is stressed in the title, and from the lecture of the text the reader gets the impression that the main message is performance improvement by moving from tunnel emitters to Schottky emitters. Personally, I would not put too much weight to these two aspects, for the following two reasons:

- First, the base transition time is already negligible in a GBT due to extremely narrow base formed by a single graphene sheet (and the authors note this in the manuscript, summarily ignoring this time in their analysis). In fact, the potential mismatch between the emitter and graphene making the electrons in graphene hot may be here more a hindrance than a help, because it increases the quantum scattering of electrons back to the emitter. So I would treat the role of hot electrons with care, not over-stressing their beneficial contribution to the high-frequency performance of the device.

- Second, the difficulties posed by the tunnelling process have been recognized already in the first GBT publication [25], where the prediction of the cut-off frequency of about 1.5 THz was done assuming the emitter built from 3 nm of undoped Ge on Er₂Ge₃ electrode (though lines 43-45 and 197-198 of the manuscript properly comment on the related problem with the capacitance). Schottky barriers in the emitter and collector electrodes have been afterwards analysed in 2013 by theorists from Bologna: Di Lecce, IEEE Trans. Electr. Dev. Lett. 60, 3584 (2013) and Di-Lecce, IEEE Trans. Electr. Dev. 60, 4263 (2013); their work should most certainly be cited in the final version of the paper. The important progress reported by the submitted manuscript is in my view that such devices have finally been fabricated and the superior performance confirmed, as the authors properly note in the last sentence of the Abstract.

2. For the reasons explained above, I would recommend to put much more weight on the analysis of the achieved and prospective device parameters, in particular on the estimate of the cut-off frequency and on the perspective of this estimate being valid for useful devices working in the common base configuration. Towards this end, I propose to consider the following:

- Lines 61-78 on device fabrication. Since the fabrication involves transfer of graphene and then of a SOI membrane, it is to be expected that the surfaces of Si and possibly also of Ge are covered by a chemical oxide. Such an oxide is reduced by the metal used to make a metal-semiconductor Schottky diode, but this is not the case here, because graphene does not reduce anything. Is there any information available on the interfacial chemistry? To what extent does this oxide have an effect on high-frequency performance of the device?

- Lines 80-89 (Si emitter), with reference to Fig. 2b in the main text and to the supplementary Fig. 2. The supplementary Figure proves a clear temperature dependence typical for a Schottky emission (in contrast to a tunnelling process). But the energy barrier retrieved from Fig. 2b is as high as 1.5 eV, and this is already at 3 V bias, that is, already including Schottky barrier lowering. This is much higher than a barrier expected on a semiconductor with 1 eV energy gap, and indeed exceeds the values reported for n-Si/graphene Schottky barriers, which are in the range of 0.3 to 0.9 eV; see e.g. the review by Di Bartolomeo, Physics Reports 606, 1 (2016). So I would recommend to plot the supplementary Fig. 2 in the traditional Schottky coordinates, $\ln(J/T^2)$ vs. \sqrt{V}/T , so that the accuracy of the barrier extraction can be better appreciated. If the barrier is indeed high, could it be attributed to the presence of the interfacial oxide? The barrier estimated for the Ge collector is lower (albeit also somewhat high, given the smaller band gap of Ge); can this be attributed to less efficient oxidation of Ge?

- Lines 101-107, on the cut-off frequency. The cut-off frequency is obtained from the emitter characteristics assuming the demonstrated ballistic transport (current gain close to 1) at sufficiently high emitter-base voltages. While this is correct, please note that the fabricated device has high input and low output impedance, that is, its power gain is much lower than 1 when Si is considered

as the input and Ge as the output terminal. On the other hand, Fig. 3b indicates that reversing the role of these terminals would also lead to a working device: due to quantum capacitance effects in graphene, one can still control the Si-graphene potential difference by varying the POSITIVE bias on Ge, so that reasonable power gain can be obtained. But this rises the question of the role of the leakage current in the magnitude of this effect; the authors may want to address this more directly, which might be facilitated by providing plots showing base and collector currents and also plotting the input characteristic showing in this case I_C versus V_{CB} at various constant V_{EB} (there one can readily see, e.g., if and how much does the separation between the IV lines drawn for various V_{EB} depend on the choice of V_{CB} , that is, by how much does the collector indeed influence the emitter). How would the output characteristics (families of I_E vs V_{EB} at constant I_C) in such a configuration look like?

- Lines 191-198, on predicted THz operation, and the corresponding supplementary page. Please provide more information on the estimate, including the formula and the assumed and measured quantities that enter into it, so that one can more easily appreciate the role of the measured data and of the assumptions in this estimate. In particular, how was the ideal Si-Gr emitter capacitance at 1000 A/cm^2 calculated?

3. Some minor points:

- Line 34: references. Besides [6] and [7], also [25] should be cited at this point.

- Line 186 (Fig. 4) and the related discussion. The Figure is valid for small voltages, much lower than the several volts that must be applied between graphene and the terminals to achieve high frequency operation.

- Lines 258-262, characterization. How many working devices have been successfully fabricated and characterized? Are the reported results typical for many working devices? How reproducible is the process?

4. The paper is in general clearly and nicely written, but it should be linguistically improved in a few places. For example:

- Lines 37-38: "atomic thickness /.../ of graphene will benefit /.../ the base resistance". Actually, it degrades the base resistance, because the base is so thin. So this sentence should be split into two:

one about the thickness and the current gain, and the other about the mobility and the base resistance.

- Line 128: "Previously those scattered electrons which could not cross the collector barrier can now tunnel under the barrier". The word "previously " is superfluous or misplaced (BTW, "which" should be preceded by a comma).

Reviewer #3 (Remarks to the Author):

The paper describes a novel silicon-grapheme-germanium hot electron transistor with record performance. Before the paper can be accepted, the following issues need to be addressed:

1/On p. 4 it is stated that with a positive voltage applied to the emitter junction, electrons are emitted... In my opinion, this statement is wrong or at least not carefully formulated.

Electrons will be emitted from the emitter when a negative bias is applied to it.(see also Fig. 2a).

2/ The barrier of 1.47 eV seems to be rather high in my opinion. How do we have to interpret this in terms of electron affinity and work function of silicon and grapheme? In addition, barrier seems to be dropping at higher temperatures. Can you explain this degradation of the leakage current for higher temperatures?

3/ The Authors state that the depletion in the silicon emitter can easily reach several micrometers. However, in the present structure, with a thickness of 880 nm, the silicon emitter is in my opinion, for the quoted doping density, fully depleted. This should also modify the band structure of the transistor in Fig. 1e. It has perhaps also consequences for the barrier height.

4/ I do not understand the statement on p. 12 that ohmic contacts were obtained due to the generation current at the etched surface. It is also not clear from the supplementary material (Fig. 1). In Fig. 1c, one can indeed observed that after prolonged etching (7 min) ohmic behavior is found. But I do not agree with the explanation or at least I do not understand it. It is known that it is easier

to make a good ohmic contact to a damaged surface, by defect-assisted leakage, so this could be the reason. At the same time, it is known that dry etching damage can lead to the introduction of shallow donors, so that the doping density can be changed. By the way, I presume that the silicon membrane is p-type (the SOI film?).

I would suggest to verify the resistivity of the membrane if possible to check whether there are some changes in the net doping density. This will also have a drastic impact on the fully or partially depleted nature of the 880 nm film.

minor detail:

p. 2 supplementary line 33: can now tunnel through the collector barrier to be collected. This is a strange sentence - perhaps to be collected can be removed or replaced.

Responses to the comments of reviewers

Reviewer #1:

The authors fabricated a graphene-base transistor with Si emitter and Ge collector. The main claims of the paper are record current density and potential for THz operation. However, I find that the claims are not well supported by the experimental data and are incorrect in some cases. The manuscript is not suitable for publication in the current format. Here are my comments,

Response:

Thank you very much for your review. In accordance with the reviewer's valuable and insightful comments, we have carefully revised our manuscript and addressed all the concerns of the reviewer in the revised manuscript. Details are discussed as follows.

1. The authors experimentally demonstrated the graphene-base heterojunction transistor proposed by Di Lecce et al. (IEEE TRANSACTIONS ON ELECTRON DEVICES, VOL. 60, NO. 12, DECEMBER 2013). It would be useful to discuss the performance gap between the theoretical results proposed by Di Lecce et al. and the experimental results presented in this paper.

Response:

As the reviewer pointed out, a pioneering theoretical study on graphene-base heterojunction transistors (GBHT) has been done with a device structure of silicon-graphene-silicon by Di Lecce et al¹. In our study, we have demonstrated the first vertical silicon-graphene-germanium (Si-Gr-Ge) transistor experimentally. In the theoretical

study of Di Lecce et al.¹, using neglected contact resistance and ideal silicon-graphene interface, as well as heavily-doped silicon emitter ($3 \times 10^{19} \text{ cm}^{-3}$) and collector (10^{18} or $3 \times 10^{18} \text{ cm}^{-3}$), the THz operation was expected when the collector current was larger than 10^6 A/cm^2 . Indeed, there exists a performance gap between the theoretical results and our experimental results, which is mainly induced by the key factors of contact and interface quality, as well as emitter doping and collector selection. Based on our experimental results, an ideal case, ignoring the series resistance, with an ideal interface was considered for a transistor with a heavily-doped n^+ -Si-Gr emitter and a thin collector, and the THz operation is expected with an emitter current of about $4.7 \times 10^6 \text{ A/cm}^2$, which is consistent with the theoretical predictions¹ reported by Di Lecce et al. The details are presented as below.

As shown in **Fig. R1a**, the experimental data (black solid line) of the I-V characteristics of the Si-Gr emitter was fitted using a Schottky junction model without considering series resistance $J=J_0[\exp(V/\eta V_t)-1]$, where the fitted leakage current density is $J_0=3.3 \times 10^{-5} \text{ A/cm}^2$, the ideality factor $\eta=1.85$ and V_t is the thermal voltage (red dashed line). When the voltage is larger than 0.2 V, the model deviates from the experimental data because of the series resistance. An ideal case was considered. The series resistance was ignored and an ideal interface with unit ideality factor $\eta=1$ was assumed (blue dash-dot line), which further increases the current and conductance.

Fig. R1 Terahertz operation using the Si-Gr emitter in an ideal case. **a** The I-V characteristics of the Si-Gr emitter obtained from experiments (black solid line), and those fitted using a Schottky junction model without considering series resistance (red dashed line) and further using an ideality factor of 1 (blue dash-dot line). **b** When a heavily-doped Si emitter and a thin collector are used, the ideal Si-Gr emitter without series resistance and with ideal interface shows a cut-off frequency of 1.06 THz at 4.7×10^6 A/cm².

A heavily-doped n⁺-Si emitter was used in this estimation (doping concentration: 8×10^{19} cm⁻³). The I-V characteristics can be predicted by the model of the ideal lightly-doped Si emitter case (blue dash-dot line in **Fig. R1a**), if the same Schottky barrier height $q\phi_{Bn} = 0.64$ eV at the graphene side is assumed which is determined by experiments (see **Fig. R2**). The conductance g_m can thus be estimated from the I-V characteristics. The capacitance C_e was calculated from the plate-capacitor model as $C_e = \epsilon \epsilon_0 / W$, $W = (2 \epsilon \epsilon_0 / q N_d (\psi_{bi} - V - V_t))^{0.5}$, $\psi_{bi} = \phi_{Bn} + V_n$, and $V_n = V_t (\ln(N_d / N_c) + 2^{-3/2} \cdot (N_d / N_c))$, where ϵ is the relative dielectric constant, ϵ_0 is the permittivity in vacuum, W is the width of the depletion width for Si, q is elementary charge, N_d is the doping concentration of Si of 8×10^{19} cm⁻³, $q\psi_{bi}$ is the potential barrier height at the Si side when no bias is applied, V_t is the thermal voltage, $q\phi_{Bn}$ is the Schottky barrier height of about 0.64 eV, V_n is the Fermi potential of the Si, and N_c is the effective density of states in conduction band of Si. The emitter charging time τ_e can thus be calculated as $\tau_e = C_e / g_m$.

A small collector delay time τ_c can be achieved by heavily-doped semiconductors¹ or thin collector semiconductors such as multilayer 2D materials^{2,3}. Assuming a thickness of $\chi=5$ nm and a saturation velocity⁴ $v=4\times 10^6$ cm/s gives the collector delay time¹⁹ $\tau_c=\chi/(2v)=6.25 \times 10^{-14}$ s. The alpha cut-off frequency f_α was estimated by $f_\alpha=1/(2\pi(\tau_e+\tau_c))$ with ignored base transit time τ_b . With a current of about 4.7×10^6 A/cm², f_α of about 1.06 THz is obtained (**Fig. R1b**), which is consistent with the theoretical predictions, demonstrating that the graphene-base heterojunction transistor has a potential application in the THz operation¹.

Overall, the gap between theoretical and experimental results can be improved by contact engineering (such as using heavily-doped semiconductor and ion implantation) and interface engineering (Supplementary Discussion 1). A heavily-doped emitter is needed for higher current and conductance, and a heavily-doped collector or a thin 2D material collector should also be used to reduce the collector delay time.

Corresponding modification has been made in Page 1 (Lines 12-15), Page 3 (Lines 4-11) and Page 13 (Lines 5-9), all in red color in the revised manuscript as well as Supplementary Fig. 14 in the revised Supplementary Information.

2. Graphene/Si Schottky diode is a widely studied structure (Di Bartolomeo, Physics Reports Vol. 606, Jan 2016, Pp 1-58). Can the authors benchmark their Schottky diode against the literature?

Response:

We thank the reviewer very much for kind suggestion. Compared with the previous Gr-on-Si junctions, our work first reported the Si-membrane-on-Gr junction. The

ideality factor and the Schottky barrier height of our Si-Gr junction are consistent with the results of previous Gr-on-Si junctions reported in literature⁵⁻¹³ as listed in **Table R1**.

Table R1 Comparison of Schottky barrier height ($q\phi_{Bn}$) and ideality factor of graphene/n-Si Schottky diodes.

Reference	$q\phi_{Bn}$ (eV)	Ideality factor	Si doping	Notes
[5]	0.86	1.2-5	$2-6 \times 10^{15} \text{ cm}^{-3}$	-
[6]	0.62	1.08	$1 \times 10^{16} \text{ cm}^{-3}$	CVD graphene on etched Cu
[6]	0.57	1.5	$1 \times 10^{16} \text{ cm}^{-3}$	-
[7]	0.69	1.46	$1 \times 10^{15} \text{ cm}^{-3}$	exfoliated graphene
[7]	0.83	2.53	$1 \times 10^{15} \text{ cm}^{-3}$	CVD graphene
[8]	0.79	1.6-2	$0.8-1 \times 10^{15} \text{ cm}^{-3}$	-
[8]	0.89	1.3-1.5	$0.8-1 \times 10^{15} \text{ cm}^{-3}$	Graphene doped with TFSA
[9]	0.833-0.858	1.56-1.58	$0.05-0.2 \text{ } \Omega \times \text{cm}$	-
[10]	0.79	1.41	$5 \times 10^{14} \text{ cm}^{-3}$	-
[11]	0.71	3.7	$1-10 \text{ } \Omega \times \text{cm}$	-
[12]	0.407	1.1	$1 \times 10^{16} \text{ cm}^{-3}$	-
This work	0.64	1.85	$1 \times 10^{15} \text{ cm}^{-3}$	-

In order to determine the Schottky barrier height, a Si-Gr junction with Au electrodes was fabricated and the traditional temperature-dependent I-V measurements of the forward current were carried out (**Fig. R2a**). The currents at a small forward bias of 0.1 V (−0.1 V at n-Si side) were considered to avoid the effect of series resistance. The following model was used to fit the relationship between current and temperature: $\ln(I/T^2) = C - q(\phi_{Bn} - V/\eta)/k \cdot (1/T)$, where I is current, T is temperature, C is a constant, q is elementary charge, $q\phi_{Bn}$ is Schottky barrier height, V is voltage, η is the ideality factor of 1.85, and k is the Boltzmann constant. $q\phi_{Bn}$ was fitted to be ~0.64 eV at room

temperature (**Fig. R2b**). As temperature decreases, the thermal emission becomes weaker and the tunnel current starts to emerge which shows a weak temperature dependence.

Fig. R2 Temperature-dependent I-V characteristics of the Si-Gr junction. **a** The temperature dependence of the current indicates a Schottky behavior. The temperatures are 251.2, 261.7, 273.2, 285.6 and 299.3 K. **b** An Arrhenius plot at a voltage of -0.1 V gives a Schottky barrier height of 0.64 eV at room temperature.

Corresponding modification has been made in Page 6 (Lines 5-9, in red color) and Fig. 2b in the revised manuscript as well as Supplementary Fig. 3 and Supplementary Table 1 in the revised Supplementary Information.

3. I'm not convinced that the contact between probe and Si is ohmic. The resistivity of the Si is too high (6 ohm.cm) to make ohmic contact just by landing probes on it. The I-V presented in the supplementary is linear but current level is too low compared to graphene if I assume same area. The authors need to fabricate proper ohmic contact to study the device operation. Otherwise, the device operation is limited by the contact resistance between probe and Si.

Response:

Thank you for your valuable suggestion. According to this suggestion, we fabricated Au electrodes after the surface of Si was etched using RIE (**Fig. R3**). A

transfer length method test¹⁴ was carried out to investigate the Ohmic contact between Au and RIE-etched Si surface (**Fig. R4a**). The I-V characteristics between adjacent Au electrodes indicate an Ohmic contact (**Fig. R4b**). More details are discussed as follows.

The resistance and distance have a linear relationship as fitted (dash line in **Fig. R4c**). The slope equals to $\rho/(Wt)$, where ρ is the resistivity of Si, W is the width of Au electrode (93 μm), and t is the thickness of the Si trip (about 880 nm) (**Supplementary Fig. 15**). ρ was fitted to be 4.5 $\Omega\cdot\text{cm}$, which is similar to the original SOI wafer (1-6 $\Omega\cdot\text{cm}$), indicating that the RIE processes does not change the doping concentration of Si obviously. The intercept on Y-axis (when distance becomes 0) equals to $2R_c$ where R_c is the contact resistance between Au and the Si strip (1.3 k Ω). The intercept on X-axis (when resistance becomes 0) equals to $-2L_t$, where L_t is the transfer length. The specific contact resistivity ρ_c was calculated as $\rho_c = L_t^2 \cdot (\rho/t) = 2.72 \times 10^{-3} \Omega\cdot\text{cm}^2$. With the area of the Si membrane ($S = 6.76 \times 10^{-6} \text{ cm}^2$), the contact resistance between Au and the Si membrane in the transistor was estimated to be $\rho_c/S = 402 \Omega$, which is consistent with the total series resistance ($\sim 426 \Omega$) for the Si-Gr emitter at -5 V (**Fig. 2a**). This contact resistance leads to an improved on-current of 692 A/cm^2 at -5 V , compared with 470 A/cm^2 at -10 V without Au electrode.

Fig. R3 Device design and fabrication. **a** A Si-Gr-Ge transistor is built by directly stacking a Si membrane, single-layer graphene and a Ge substrate. **b** Optical image of a Si-Gr-Ge transistor (scale bar: 20 μm). **c** SEM image of a Si membrane on graphene (scale bar: 4 μm). **d** Illustration of the cross section of the transistor. **e** Illustration of the basic operating principle of the transistor.

Fig. R4 Resistivity measurements of the RIE-etched Si layer on an SOI wafer based on a transfer length method. **a** Optical micrograph of a Si strip with deposited Au electrodes in different distances of 8, 12, 17, 23 and 30 μm. **b** I-V characteristics between adjacent Au electrodes. **c** Relationship of the resistance and distance.

As the reviewer pointed out, the previous current is relatively small. This is because that the Si is lightly-doped and the current conduction route has a long distance and a small cross section when the surface of the top Si layer of the SOI wafer is examined by probes directly.

Corresponding modification has been made in Page 4 (Lines 7-9), Page 5 (Lines 8-10), Page 14 (Lines 8-11), all in red color and Figs. 1 and 2a in the revised manuscript as well as Supplementary Fig. 2 in the revised Supplementary Information.

4. Authors claim that they have achieved record current density. However, from Fig. 2(c) it is obvious that current work has similar or lower current when compared with ref 17 for the same applied bias and it can achieve higher current density only when the applied bias is larger. Moreover, Guo et al. (IEEE Electron Device Letters,

Vol.39 , No. 4 , pp. 634-637, April 2018) have demonstrated similar current density ($\sim 233 \text{ A/cm}^2$) with much smaller applied bias across tunnel emitter. Could the authors please comment on that?

Response:

Thank you for the valuable comments. Indeed, the tunnel emitter mentioned by the reviewer showed a larger current at small voltages^{2,3}. However, the on-current of our Schottky emitter can now reach 692 A/cm^2 at -5 V which is the largest reported current value compared with the tunneling counterparts^{2,3}. According to the reviewer's suggestion, we will not emphasize on the "largest" current, and only state the fact that a current of 692 A/cm^2 is achieved at -5 V .

Besides the large current, our Schottky junction also has a small capacitance ($\sim 28 \text{ nF/cm}^2$) since Si is lightly-doped. In contrast, the capacitances for the tunnel emitters are at least one-order-of magnitude higher ($\sim 325 \text{ nF/cm}^2$ for 13.6 nm h-BN^2 and $\sim 2.66 \mu\text{F/cm}^2$ for 3 nm AlN^3). The alpha cut-off frequency f_α is not only proportional to the emitter current and conductance, but also inversely proportional to the emitter capacitance. Our Schottky emitter has an advantage in f_α at any current value compared with the tunnel emitter³. f_α is estimated to be above 1 GHz for our Schottky emitter, while it is less than 1 MHz for the tunnel emitter³ (**Fig. 2d**).

Corresponding modification has been made in Page 1 (Lines 17-18), Page 3 (Line 13), Page 6 (Lines 16-17), Page 7 (Lines 5-8) and Page 13 (Lines 12-13), all in blue color as well as Fig. 2cd in the revised manuscript.

5. I'm confused by the primed (V_c' , V_e') and regular notations (V_c and V_e). Could

the authors please clarify the difference? It is hard to understand the device operation without clarification.

Response:

We are sorry that the previous statement was not clear and led to misunderstanding. The “prime” is actually a “comma”, and “ V_e, V ” and “ V_c, V ” in the previous figures have been changed to “ $V_e (V)$ ” and “ $V_c (V)$ ”, respectively.

We have changed all the “comma” to “brackets” to separate a variable and its unit in all the figures in the revised manuscript and supplementary information.

6. How does the base-collector cold electron leakage affect the gain of the device? Did the authors consider that while calculating the gain in Fig 3c? Is the diode characteristics presented in Supp. Fig. 3b is from the same device of Fig. 3b? Please show the base current in Fig 3b. I_e and I_c should be swapped in Fig 3d.

Response:

Thank you very much for the valuable comments. According to this suggestion, we have added current component discussions, more electrical characteristics and other explanations, which have improved the manuscript significantly. The details are discussed as follows.

Concerning the current gain, the current components are illustrated in **Fig. R5**. Graphene (Gr) is connected to ground. I_e is emitted from the Si-Gr emitter. Part of I_e is collected at the Ge collector forming the effective collector current I_c' , while the other part flows to ground forming I_b' . A leakage current I_{leak} of the collector junction also contributes to the collector current I_c and base current I_b , so that $I_c = I_c' + I_{leak}$, $I_b = I_b' - I_{leak}$.

If the gain α is calculated without removing the effect of I_{leak} , it will be overestimated.

We calculated the gain after removing the effect of I_{leak} as $\alpha = I_c' / I_e = (I_c - I_{\text{leak}}) / I_e$ and

$I_{\text{leak}} = I_c$ ($V_e = 0$).

Fig. R5 Illustration of the voltage bias and current components for the transistor in the common base mode.

Concerning the diode characteristics, the electrical characteristics of the Si-Gr-Ge transistors is updated. **Fig. R6a-d** and **R6e-h** show the performances of the transistors using a lightly-doped n-Ge collector (resistivity: $\sim 1 \Omega\text{cm}$) and a heavily-doped n^+ -Ge collector (resistivity: $\sim 0.1 \Omega\text{cm}$), respectively. **Figs. R6a** and **R6e** show the I-V characteristics of the diodes of the two kinds of transistors, respectively.

Fig. R6 Electrical characteristics of the Si-Gr-Ge transistors in the common base mode. The figures in left column (**a-d**) are for the transistor using a lightly-doped n-Ge collector, and those in the right column (**e-h**) are for a heavily-doped n⁺-Ge collector. **a** The I-V characteristics of the Si-Gr and Gr-n-Ge junctions. **b** Input (I_e - V_e) and transfer (I_c - V_c) characteristics where V_c changes from 0 to 4 V. **c** Transfer (I_c' - V_e) characteristics after eliminating the influence of the collector junction leakage. Inset: common base current gain. **d** Output (I_c - V_c) characteristics. I_e changes from 100 to 500 μ A. **e-h** Corresponding electrical characteristics for the transistor using a heavily-doped n⁺-Ge

collector.

Concerning the base current, current components including the base current I_b and the effective base current I_b' of the transistor with an n^+ -Ge collector are shown in **Fig. R7**. As V_e increases from 0 to -5 V, I_e increases, and I_c' and I_b' increases correspondingly. $I_c = I_c' + I_{\text{leak}}$ also increases, where I_{leak} is the leakage current at the collector junction. $I_b = I_b' - I_{\text{leak}}$ tends to change the current direction (from <0 to >0) as I_b' increases and surpasses I_{leak} . A current $I > 0$ indicates that the current flows into the device and $I < 0$ indicates that the current flows out of the device.

Fig. R7 Current components in the transistor with an n^+ -Ge collector in the common base mode: base current I_b with corresponding emitter current I_e and collector current I_c in **a** the logarithmic coordinates and **b** the linear coordinates, and effective base current I_b' with corresponding emitter current I_e and effective collector current I_c' in **c** the logarithmic coordinates and **d** the linear coordinates. Data for different collector bias V_c is denoted by different colors: black, red, blue, pink and green are for $V_c = 0, 1, 2, 3, 4$ V respectively.

Concerning the symbol for current definition, **Fig. R8** shows I-V characteristics

for the transistor with a heavily-doped germanium, in which the emitter and collector are exchanged so that electrons are emitted from n^+ -Ge ($V_c < 0$) and collected at Si ($V_e > 0$). In this paper, the current and voltage at the Ge (Si) side is denoted as V_c (V_e) and I_c (I_e), respectively, to avoid misunderstanding (especially considering a “reverse working mode” shown in Supplementary Figs. 12, 13).

Fig. R8 I-V characteristics of the transistor with a heavily-doped germanium, in which the emitter and collector are exchanged. Electrons are emitted from n^+ -Ge ($V_c < 0$) and collected at Si ($V_e > 0$).

Corresponding modification has been made in Page 8 (Lines 12-20) and Page 9 (Lines 4-12) in blue color, and Page 9 (Lines 19-20 in pink color), and Fig. 3 in the revised manuscript as well as Supplementary Figs. 5, 9, 10 in the revised Supplementary Information.

7. The authors mentioned that their device can potentially overcome the frequency limitation of BJT and predicted THz operation for optimized device geometry without considering any non-idealities. InP HBT technologies have already reached $f_t > 0.5$ THz and $f_{max} > 1$ THz (M. Urteaga et al. 69th Device Research Conference, 2011). Could the authors please compare their projection with the InP HBT experimental results before claiming improvement over the state-of-the-art?

Response:

Thanks for the valuable comment. As pointed out by the reviewer, InP HBT technologies have already achieved great progress toward THz operation¹⁵⁻¹⁸. In this study, the terahertz operation using the Si-Gr emitter in an ideal case was estimated based on the experimental results (**Fig. R1**). When the current is larger than 4.7×10^6 A/cm², the alpha cut-off frequency f_α can exceed 1 THz, thanks to the negligible base transit time. If a current gain approaching 1 is assumed, f_T is above 1 THz, which is larger than the experimental result of an InP HBT. As suggested by Di Lecce et al.¹, the cut-off frequency of an HBT is ultimately limited by the transit time in the base, while a graphene-base heterojunction transistor has no such limitation. This is a key advantage over the HBT, however further effort is needed for its development.

Corresponding modification has been made in Page 2 (Lines 14-15 in red color) in the revised manuscript as well as in Supplementary Fig. 14 in the revised Supplementary Information.

8. The authors also mentioned that it would require 1 μm thick emitter to achieve THz operation. Would it increase the emitter transit time?

Response:

Thanks for the valuable comment. Because the current mechanism of the emitter junction is not the drift mechanism, the emitter transit time is usually ignored and not mentioned^{19,20}. For the collector junction in a normal n-p-n BJT, when it is reversely biased, an electron must go through the space charge region of the collector by drift, and a “transit time” is needed. However, for the emitter junction, when it is forwardly

biased, the carriers go through the space charge region by diffusion. When the forward bias has a slight change, the delay time mainly comes from the time for the free carriers to charge the space charge region which is mainly τ_e as discussed in the main text. After that charging time, a carrier does not need to go through the whole emitter space charge region to reach the base, but only by diffusion from interface between the space charge region and the base. Similar analysis can be made to a Schottky emitter. After the charging time τ_e , an electron does not need to travel the whole space charge region to enter the base, but only by emission at the interface between the semiconductor and metal (or graphene), leading to a negligible emitter transit time. On the other hand, as the simulation work by Di Lecce et al.¹, a heavily-doped n⁺-Si emitter (doping concentration: $8 \times 10^{19} \text{ cm}^{-3}$) now is used to estimate the THz operation (**Fig. R1**), and the space charge region is in the nanometer scale.

Corresponding modification has been made in Supplementary Fig. 14 in the revised Supplementary Information.

Reviewer #2: The paper reports on a successful fabrication of vertical silicon-graphene-germanium transistors with performance that notably surpasses other results published so far. Based on the measured data, the authors predict for such devices high-frequency operation with cut-off frequency of 1 THz at 1 kA/cm², that is, at current levels comparable to those achieved by the devices they made. This makes the concept of a graphene base transistor (GBT), i.e., of a transistor in which the input-output current flows in the direction normal to the graphene sheet, a realistic alternative to the

usually studied concept of a graphene field-effect transistor, i.e., of a transistor in which the input-output current flows in the plane of the graphene sheet. This achievement is therefore of sufficient novelty and of sufficient general interest to justify the publication of the report in Nature Communications. Nevertheless, my opinion is that the distribution of accents in the main message as well as in the write-up should be optimized. The paper would also benefit from some improvements in the presentation of the acquired data and in the discussion. There are several points to be addressed, so I recommend publication of the paper after a major revision.

Response:

Thank you very much for your positive comments.

1. The importance of the hot-electron character of the device is stressed in the title, and from the lecture of the text the reader gets the impression that the main message is performance improvement by moving from tunnel emitters to Schottky emitters. Personally, I would not put too much weight to these two aspects, for the following two reasons: First, the base transition time is already negligible in a GBT due to extremely narrow base formed by a single graphene sheet (and the authors note this in the manuscript, summarily ignoring this time in their analysis). In fact, the potential mismatch between the emitter and graphene making the electrons in graphene hot may be her more a hindrance than a help, because it increases the quantum scattering of electrons back to the emitter. So I would treat the role of hot electrons with care, not over-stressing their beneficial contribution to the high-frequency performance of the device. Second, the difficulties posed by the tunneling process have been recognized

already in the first GBT publication [25], where the prediction of the cut-off frequency of about 1.5 THz was done assuming the emitter built from 3 nm of undoped Ge on Er₂Ge₃ electrode (though lines 43-45 and 197-198 of the manuscript properly comment on the related problem with the capacitance). Schottky barriers in the emitter and collector electrodes have been afterwards analyzed in 2013 by theorists from Bologna: Di Lecce, IEEE Trans. Electr. Dev. Lett. 60, 3584 (2013) and Di-Lecce, IEEE Trans. Electr. Dev. 60, 4263 (2013); their work should most certainly be cited in the final version of the paper. The important progress reported by the submitted manuscript is in my view that such devices have finally been fabricated and the superior performance confirmed, as the authors properly note in the last sentence of the Abstract.

Response:

Thanks for your valuable comments. As the reviewer pointed out, graphene itself is thin enough to give a negligible base transition time for a GBT, and this work can be considered as the first experimental demonstration of the theoretical work by Di Lecce et al¹. According to this suggestion, the title of the paper has been changed to “A vertical silicon-graphene-germanium transistor”. In the revised manuscript, in the introduction of a graphene-base transistor, it is emphasized that the graphene base is almost transparent to the electron transport leading to a negligible τ_b because of the atomic thickness.

According to the reviewer’s suggestion, a main claim of the paper has been modified as that the graphene-base heterojunction transistor (GBHT) theoretically proposed by Di Lecce et al. has been experimentally realized in a silicon-graphene-

germanium transistor, and its potential for THz operation has been confirmed based on experiments and modeling. In the revised introduction, it is stressed that pioneering theoretical study on the GBHT has been done with a device structure of silicon-graphene-silicon^{1,21}, and theoretically the THz operation can be realized when the collector current is larger than 10^6 A/cm². One of main contributions of our work is that we have demonstrated the first vertical silicon-graphene-germanium (Si-Gr-Ge) transistor, and with further engineering, the vertical semiconductor-graphene-semiconductor transistor is expected to be one of the most promising devices for ultra-high frequency operation.

Corresponding modification has been made in the title and Page 1 (Lines 12-15 in red color), Page 2 (Lines 21-22) and Page 3 (Lines 1-3) in pink color, and Page 3 (Lines 4-11 in red color) in the revised manuscript.

2. For the reasons explained above, I would recommend to put much more weight on the analysis of the achieved and prospective device parameters, in particular on the estimate of the cut-off frequency and on the perspective of this estimate being valid for useful devices working in the common base configuration. Towards this end, I propose to consider the following: Lines 61-78 on device fabrication. Since the fabrication involves transfer of graphene and then of a SOI membrane, it is to be expected that the surfaces of Si and possibly also of Ge are covered by a chemical oxide. Such an oxide is reduced by the metal used to make a metal-semiconductor Schottky diode, but this is not the case here, because graphene does not reduce anything. Is there any information available on the interfacial chemistry? To what extent does this oxide have an effect on

high-frequency performance of the device?

Response:

Thanks for the valuable comments. As the reviewer pointed out, the quality of the interface may influence the performance of the device. In this study, the Si membrane and Ge substrate were made by HF-based process, and their surfaces are hydrogen-terminated. The XPS analyses were carried out to examine the surfaces of Si and Ge that were just cleaned by HF, and no obvious oxide was detected. This result indicates that such surfaces should be free of oxide even a transfer process is used, which is consistent with that reported by Kiefer et al.²²

On the other hand, the surface of the transferred graphene is contaminated by PMMA residue as reported previously^{23,24}. This is expected to influence the series resistance and ideality factor of the emitter junction, and thus the on-current and high-frequency performance of the transistor. The strategy to improve the quality of the interface is discussed in **Supplementary Discussion 1**.

Corresponding modification has been made in Supplementary Fig. 14.

- Lines 80-89 (Si emitter), with reference to Fig. 2b in the main text and to the supplementary Fig. 2. The supplementary Figure proves a clear temperature dependence typical for a Schottky emission (in contrast to a tunneling process). But the energy barrier retrieved from Fig. 2b is as high as 1.5 eV, and this is already at 3 V bias, that is, already including Schottky barrier lowering. This is much higher than a barrier expected on a semiconductor with 1 eV energy gap, and indeed exceeds the values reported for n-Si/graphene Schottky barriers, which are in the range of 0.3 to 0.9 eV;

see e.g. the review by Di Bartolomeo, Physics Reports 606, 1 (2016). So I would recommend to plot the supplementary Fig. 2 in the traditional Schottky coordinates, $\ln(J/T^2)$ vs. \sqrt{V}/T , so that the accuracy of the barrier extraction can be better appreciated. If the barrier is indeed high, could it be attributed to the presence of the interfacial oxide? The barrier estimated for the Ge collector is lower (albeit also somewhat high, given the smaller band gap of Ge); can this be attributed to less efficient oxidation of Ge?

Response:

Thanks very much for your valuable comment. According to the reviewer's suggestion, we made new Si-graphene junctions with Au electrodes (**Fig. 1**), and traditional temperature-dependent I-V measurements of the forward current were carried out to determine the Schottky barrier height. To fit the Schottky barrier height, currents at a small forward bias of 0.1 V (−0.1 V at n-Si side) were considered to avoid the effect of series resistance (**Fig. R9a**). The following model was used to fit relationship between current and temperature¹⁹: $\ln(I/T^2)=C-q(\phi_{Bn}-V/\eta)/k\cdot(1/T)$, where I is current, T is temperature, C is a constant, q is elementary charge, $q\phi_{Bn}$ is Schottky barrier height, V is voltage, η is the ideality factor of 1.85, and k is the Boltzmann constant. $q\phi_{Bn}$ was fitted to be ~0.64 eV at room temperature (**Fig. R9b**). As the temperature decreases, the thermal emission becomes weaker and the tunnel current starts to emerge which shows a weak temperature dependence. The value of $q\phi_{Bn}$ is consistent with the reported results (0.3-0.9 eV)¹³, demonstrating that no obvious oxide exists on the surfaces of Si and Ge as discussed above. The value of $q\phi_{Bn}$ is different

from the previous result which was obtained from the analysis of the leakage current. It is known that the leakage current is easily affected by the edge current¹⁹, which may result in the difference. The Schottky barrier height of Gr-Ge junction was also evaluated using the above method, and the obtained $q\phi_{\text{Bn}}$ is ~ 0.26 eV (**Fig. R10**).

Fig. R9 Temperature-dependent I-V characteristics of the Si-Gr junction. **a** The temperature dependence indicating a Schottky behavior. The temperatures are 251.2, 261.7, 273.2, 285.6, and 299.3 K. **b** An Arrhenius plot at a voltage of -0.1 V gives a Schottky barrier height of 0.64 eV at room temperature.

Fig. R10 Temperature-dependent I-V characteristics of a Gr-n-Ge Schottky junction. **a** I-V characteristics at the temperatures of 251.2, 261.7, 273.2, 285.6, and 299.3 K. **b** The Schottky barrier height was fitted to be $q\phi_{\text{Bn}} = 0.26$ eV at a small forward bias of 0.1 V (-0.1 V at n-Ge side) which is lower than that of the Si-Gr emitter junction.

Corresponding modification has been made in Page 6 (Lines 5-9) and Fig 2b in the revised manuscript as well as Supplementary Fig. 3 and Supplementary Fig. 6 in the revised Supplementary Information.

- Lines 101-107, on the cut-off frequency. The cut-off frequency is obtained from

the emitter characteristics assuming the demonstrated ballistic transport (current gain close to 1) at sufficiently high emitter-base voltages. While this is correct, please note that the fabricated device has high input and low output impedance, that is, its power gain is much lower than 1 when Si is considered as the input and Ge as the output terminal. On the other hand, Fig. 3b indicates that reversing the role of these terminals would also lead to a working device: due to quantum capacitance effects in graphene, one can still control the Si-graphene potential difference by varying the POSITIVE bias on Ge, so that reasonable power gain can be obtained. But this rises the question of the role of the leakage current in the magnitude of this effect; the authors may want to address this more directly, which might be facilitated by providing plots showing base and collector currents and also plotting the input characteristic showing in this case I_C versus V_{CB} at various constant V_{EB} (there one can readily see, e.g., if and how much does the separation between the IV lines drawn for various V_{EB} depend on the choice of V_{CB} , that is, by how much does the collector indeed influence the emitter). How would the output characteristics (families of I_E vs V_{EB} at constant I_C) in such a configuration look like?

Response:

As the reviewer pointed out, there was a power gain issue of the previous transistor with a heavily-doped n^+ Ge collector (resistivity: $\sim 0.01 \Omega\text{cm}$), and a “reverse working mode” was suggested. We thank the reviewer very much for the valuable comments and suggestions. According to this comment, first we used a new n^+ -Ge collector (resistivity: $\sim 0.1 \Omega\text{cm}$) to rebuild the transistor, and a working region with power gain

larger than 1 can be found. The electrical characteristics of the Si-Gr-Ge transistors were updated accordingly. **Figure R11a-d** and **e-h** show the performances of the transistor using a lightly-doped n-Ge collector (resistivity: $\sim 1 \Omega\text{cm}$) and a heavily-doped n^+ -Ge collector (resistivity: $\sim 0.1 \Omega\text{cm}$), respectively.

Fig. R11 Electrical characteristics of the Si-Gr-Ge transistors in the common base mode. The figures in left column (**a-d**) are for the transistor using a lightly-doped n-Ge collector, and those in the right column (**e-h**) are for a heavily-doped n⁺-Ge collector. **a** The I-V characteristics of the Si-Gr and Gr-n-Ge junctions. **b** Input (I_e - V_e) and transfer (I_c - V_e) characteristics where V_c changes from 0 to 4 V. **c** Transfer (I_c '- V_e) characteristics after eliminating the influence of the collector junction leakage. Inset: common base current gain. **d** Output (I_c - V_c) characteristics. I_e changes from 100 to 500 μ A. **e-h** Corresponding electrical characteristics for the transistor using a heavily-doped n⁺-Ge collector.

The input conductance $g_e=dI_e/dV_e$ as well as the current gain $\alpha=dI_c/dI_e$ can be calculated from the input characteristics (**Fig. R11f**). For the transistor with a heavily-doped n⁺-Ge collector, the output characteristics with different voltages V_e are also shown in **Fig. R12**, from which the output conductance $g_c=dI_c/dV_c$ at different voltages V_e can be calculated. The power gain can thus be expressed as $A_p=(dI_c \cdot dV_c)/(dI_e \cdot dV_e)=(dV_c/dI_e) \cdot (dI_e/dV_e) \cdot (dI_c/dI_e)^2=g_e/g_c \cdot \alpha^2$. When $V_c=3$ V and $V_e=-5$ V, the power gain $A_p=1.5$; and when $V_c=3$ V and $V_e=-4$ V, the power gain $A_p=1.6$. Therefore, the region with power gain larger than 1 can be found.

Fig. R12 Output characteristics (I_c - V_c) of the transistor with a heavily-doped n⁺-Ge collector with V_e from 0 to -5 V.

Next, we explored the reverse working mode (Ge was used as input with a bias $V_c>0$, and Si as output with a bias $V_e<0$) of the transistor with a heavily-doped n⁺-Ge collector. The transistor was in the common base mode. Graphene was connected to ground. As shown in the input (I_c - V_c) and transfer (I_e - V_c) characteristics (**Fig. R13a**),

when V_c increases, I_c increases, and I_e increases accordingly because of the quantum capacitance effect of graphene. This effect can also be observed in the output characteristics (Fig. R13b). The highest current gain dI_e/dI_c is about 77% when $V_c=1.55$ V and $V_e=-4$ V.

Fig. R13 Electrical performance of the transistor with a heavily-doped n^+ -Ge collector in a reverse working mode. **a** Input (I_c - V_c) and transfer (I_e - V_c) characteristics. **b** Output (I_e - V_e) characteristics.

The base currents are shown in Fig. R14. As V_c increases from 0 to 5 V, I_c increases and I_e increases accordingly. The leakage current I_{leak} at the collector junction also increases, and $I_b = I_b' - I_{leak}$ tends to change the current direction (from >0 to <0) as I_{leak} increases and surpasses I_b' . Here I_b' is effective base current. A current $I > 0$ indicates that the current flows into the device and $I < 0$ indicates that the current flows out of the device. (Current components are illustrated in Supplementary Fig. 5.)

Fig. R14 Base current I_b with the corresponding emitter current I_e and collector current

I_c in **a** the logarithmic coordinates and **b** the linear coordinates for the transistor in the reverse working mode (Fig. R13). Data for different collector bias V_e is denoted by different colors: black, red, blue, pink and green are for $V_e=0, -1, -2, -3, -4$ V respectively.

Corresponding modification has been made in Page 9 (Lines 4-12) and Page 12 (Lines 14-16) in blue color and Fig. 3 in the revised manuscript as well as Supplementary Figs. 8, 12 and 13 in the revised Supplementary Information.

- Lines 191-198, on predicted THz operation, and the corresponding supplementary page. Please provide more information on the estimate, including the formula and the assumed and measured quantities that enter into it, so that one can more easily appreciate the role of the measured data and of the assumptions in this estimate. In particular, how was the ideal Si-Gr emitter capacitance at 1000 A/cm^2 calculated?

Response:

Thanks very much for your valuable comment. According to the reviewer's suggestion, more details about the estimation are given below. The Schottky barrier height is updated to be 0.64 V as measured. A heavily-doped n^+ -Si emitter was used in this estimation as the work of Di Lecce et al¹.

As shown in **Fig. R15a**, the experimental data (black solid line) of the I-V characteristics of the Si-Gr emitter was fitted by using a Schottky junction model without considering series resistance $J=J_0[\exp(V/\eta V_t)-1]$, where the fitted leakage current density is $J_0=3.3 \times 10^{-5} \text{ A/cm}^2$, the ideality factor $\eta=1.85$, and V_t is the thermal voltage (red dashed line). When the voltage is larger than 0.2 V, the model deviates from the experimental data because of the series resistance. An ideal case was considered. The series resistance was ignored and an ideal interface with unit ideality

factor $\eta=1$ was assumed (blue dash-dot line), which further increases the current and conductance.

Fig. R15 Terahertz operation using the Si-Gr emitter in an ideal case. **a** The I-V characteristics of the Si-Gr emitter obtained from experiments (black solid line), and those fitted using a Schottky junction model without considering series resistance (red dashed line) and further using an ideality factor of 1 (blue dash-dot line). **b** When a heavily-doped Si emitter and a thin collector are used, the ideal Si-Gr emitter without series resistance and with ideal interface shows a cut-off frequency of 1.06 THz at 4.7×10^6 A/cm².

A heavily-doped n⁺-Si emitter was used in this estimation (doping concentration: 8×10^{19} cm⁻³) and the I-V characteristics can be predicted by the model of the ideal lightly-doped Si emitter case (blue dash-dot line in **Fig. R15a**), if the same Schottky barrier height $q\phi_{Bn}=0.64$ eV at the graphene side was assumed which is determined by experiments. The conductance g_m can thus be estimated from the I-V characteristics. The capacitance C_e was calculated from the plate-capacitor model as $C_e = \epsilon\epsilon_0/W$, $W = (2\epsilon\epsilon_0/qN_d(\psi_{bi} - V - V_t))^{0.5}$, $\psi_{bi} = \phi_{Bn} + V_n$, $V_n = V_t \cdot (\ln(N_d/N_c) + 2^{-3/2} \cdot (N_d/N_c))$, where ϵ is the relative dielectric constant, ϵ_0 is the permittivity in vacuum, W is the width of the depletion width for Si, q is elementary charge, N_d is the doping concentration of Si of 8×10^{19} cm⁻³, $q\psi_{bi}$ is the potential barrier height at the Si side when no bias is applied, V_t is the thermal voltage, $q\phi_{Bn}$ is the Schottky barrier height of ~ 0.64 eV, V_n is the Fermi

potential of the Si, and N_c is the effective density of states in conduction band of Si. The emitter charging time τ_e can thus be calculated as $\tau_e=C_e/g_m$. A small collector delay time τ_c can be realized by heavily-doped semiconductors¹ or thin collector semiconductors such as multilayer 2D materials^{2,3}. Assuming a thickness of $\chi=5$ nm and a saturation velocity⁴ $v=4\times 10^6$ cm/s gives the collector delay time¹⁹ $\tau_c=\chi/(2v)=6.25\times 10^{-14}$ s. The alpha cut-off frequency f_α was estimated by $f_\alpha=1/(2\pi(\tau_e+\tau_c))$ with ignored base transit time τ_b . With a current of about 4.7×10^6 A/cm², f_α of about 1.06 THz is obtained (**Fig. R15b**), which is consistent with the theoretical predictions, demonstrating that the graphene-base heterojunction transistor has a potential application in the THz operation¹.

Since a heavily-doped Si is used to perform THz operation, the capacitance is about 19 $\mu\text{F}/\text{cm}^2$ when f_α is 1 THz with a current of 4.7×10^6 A/cm². The advantage of a Schottky emitter for THz operation over a tunnel emitter was discussed by Di Lecce et al.¹. As predicted by simulations, for a tunnel emitter to realize THz operation, the emitter potential barrier which is between the emitter metal and the emitter-to-base tunneling layer should be lower than 0.4 eV, which remains an engineering issue.

Corresponding modification has been made in Page 13 (Lines 5-9 in red color) and Supplementary Fig. 14 in the revised Supplementary Information.

3. Some minor points:

- Line 34: references. Besides [6] and [7], also [25] should be cited at this point.

Response:

In the revised manuscript, Reference [25] has been cited when the graphene-base

transistor is introduced, and the advantage of the thickness of graphene to reduce the base transit time has been discussed.

Corresponding modification has been made in Page 2 (Lines 21-22 in pink color) in the revised manuscript.

- Line 186 (Fig. 4) and the related discussion. The Figure is valid for small voltages, much lower than the several volts that must be applied between graphene and the terminals to achieve high frequency operation.

Response:

Thanks for your valuable comments. **Fig. R16** shows the energy band diagram of the Si-Gr-Ge transistor when a large bias V_{be} is applied to the emitter junction. Not that most of the bias is applied to the series resistance of the junction. The energy band diagram with a small voltage bias is used to illustrate the quantum capacitance effect in the transistor which also exists when the voltage is large (**Fig. 4**).

Fig. R16 Energy band diagram of a Si-Gr-Ge transistor when a large bias V_{be} applied to the emitter junction.

Corresponding modification has been made in Page 12 (Lines 12-14 in red color) in the revised manuscript and Supplementary Fig. 11 in the revised Supplementary Information.

- Lines 258-262, characterization. How many working devices have been successfully fabricated and characterized? Are the reported results typical for many working devices? How reproducible is the process?

Response:

According to the reviewer's comment, more details about the characterization have been given. About 10-30 working transistors were successfully fabricated and characterized for each wafer with a yield of about 10%-30%. I-V characteristics of the Si-Gr emitters of 28 Si-Gr-Ge transistors in one batch are shown in **Fig. R17**. Note that the currents for the devices basically keep the same order of magnitude at $-5V$. To improve the yield, larger and cleaner graphene without damage and SOI wafers with more uniform top Si and oxide layers in thickness should be used. Similar I-V characteristics of the Si-Gr emitters were obtained for transistors on different wafers as shown in **Figs. 3a** and **3e**, indicating a reproducible process.

Fig. R17 I-V characteristics of the Si-Gr emitters of 28 Si-Gr-Ge transistors in one batch.

Corresponding modification has been made in Page 16 (Lines 17-20 in red color) in the revised manuscript as well as in Supplementary Fig. 16 in the revised Supplementary Information.

4. The paper is in general clearly and nicely written, but it should be linguistically improved in a few places. For example:

Response:

Thank you very much for your positive comments.

- Lines 37-38: "atomic thickness /.../ of graphene will benefit /.../ the base resistance". Actually, it degrades the base resistance, because the base is so thin. So this sentence should be split into two: one about the thickness and the current gain, and the other about the mobility and the base resistance.

Response:

Thank you very much for the valuable suggestion. The corresponding sentences have been modified as below in the revised manuscript:

“Because of the atomic thickness, the graphene base is almost transparent to electron transport leading to a negligible τ_b . At the same time, the remarkably high carrier mobility of graphene will benefit the base resistance compared with a thin bulk material.”

Corresponding modification has been made in Page 2 (Lines 21-22) and Page 3 (Lines 1-3) in pink color in the revised manuscript.

- Line 128: "Previously those scattered electrons which could not cross the collector barrier can now tunnel under the barrier". The word "previously " is superfluous or misplaced (BTW, "which" should be preceded by a comma).

Response:

According to the reviewer's kind suggestions, we have changed the statement as

follows:

“At the collector junction interface, around the top of the barrier, the tunneling distance of an electron decreases dramatically. As a result, electrons which originally cannot cross the collector barrier can now tunnel through the barrier, which increases the current gain.”

Corresponding modification has been made in Page 9 (Lines 1-4 in red color) in the revised manuscript.

Reviewer #3:

The paper describes a novel silicon-grapheme-germanium hot electron transistor with record performance. Before the paper can be accepted, the following issues need to be addressed:

Response:

Thank you very much for your positive comments.

1/On p. 4 it is stated that with a positive voltage applied to the emitter junction, electrons are emitted... In my opinion, this statement is wrong or at least not carefully formulated. Electrons will be emitted from the emitter when a negative bias is applied to it. (see also Fig. 2a).

Response:

As the reviewer pointed out, the statement is not carefully formulated. The sentences have been modified as:

“When the device is turned on with a positive voltage V_{be} applied to the emitter

junction (when graphene is grounded, $V_e < 0$), electrons are emitted from the emitter, go through the emitter junction with a barrier height $q\phi_1$, the graphene base, then the collector junction with a lower barrier height $q\phi_2$, and eventually are collected by the collector.”

Corresponding modification has been made in Page 4 (Lines 12-16 in blue color) in the revised manuscript.

2/ The barrier of 1.47 eV seems to be rather high in my opinion. How do we have to interpret this in terms of electron affinity and work function of silicon and graphene? In addition, barrier seems to be dropping at higher temperatures. Can you explain this degradation of the leakage current for higher temperatures?

Response:

We thank the reviewer very much for valuable comment. More studies have been done about the measurement of the Schottky barrier height. It is well known that the leakage current is easily affected by the edge current¹⁹, which may influence the Schottky barrier height. Therefore, we have made new Si-graphene junctions with Au electrodes (**Fig. 1**), and instead of measuring the leakage current, traditional temperature-dependent I-V measurements of the forward current were carried out to determine the Schottky barrier height¹⁹. To fit the Schottky barrier height, the currents at a small forward bias of 0.1 V (−0.1 V at n-Si side) were used to avoid the effect of series resistance (**Fig. R18a**). The following model was used to fit the relationship between current and temperature: $\ln(I/T^2) = C - q(\phi_{Bn} - V/\eta)/k \cdot (1/T)$, where I is current, T is temperature, C is a constant, q is elementary charge, $q\phi_{Bn}$ is Schottky barrier height,

V is voltage, η is the ideality factor of 1.85, and k is the Boltzmann constant. $q\phi_{\text{Bn}}$ was fitted to be ~ 0.64 eV at room temperature (**Fig. R18b**).

Fig. R18 Temperature-dependent I-V characteristics of the Si-Gr junction. **a** The temperature dependence of the current indicates a Schottky behavior. The temperatures are 251.2, 261.7, 273.2, 285.6 and 299.3 K. **b** An Arrhenius plot at a voltage of -0.1 V gives a Schottky barrier height of 0.64 eV at room temperature.

As temperature T decreases (and $1000/T$ increases in **Fig. R18b**), the absolute value of the slope of the fitted data decreases and thus the corresponding calculated Schottky barrier height decreases. This phenomenon indicates that the temperature dependence of the current becomes weaker. It may be caused by the fact that, as temperature decreases, the thermal emission becomes weaker, and the tunnel current starts to emerge, which shows a weak temperature dependence¹⁹.

Corresponding modification has been made in Page 6 (Lines 5-9 in red color) and Fig 2b in the revised manuscript as well as Supplementary Fig. 3 in the revised Supplementary Information.

3/ The Authors state that the depletion in the silicon emitter can easily reach several micrometers. However, in the present structure, with a thickness of 880 nm, the silicon emitter is in my opinion, for the quoted doping density, fully depleted. This should also modify the band structure of the transistor in Fig. 1e. It has perhaps also

consequences for the barrier height.

Response:

Thank you for the valuable comments. We have carried out more analyses on the space charge region and band structure. When no bias is applied, the width of the space charge region W can be calculated as $W=(2\epsilon\epsilon_0/qN_d(\psi_{bi}-V_t))^{0.5}$, where N_d is the doping concentration of Si of about $1\times 10^{15}\text{ cm}^{-3}$, $\psi_{bi}=\phi_{Bn}-V_t\cdot\ln(N_c/N_d)$ is the potential barrier height at the Si side, V_t is the thermal voltage, and N_c is the effective density of states in conduction band of Si. As shown above, the measured Schottky barrier height $q\phi_{Bn}$ is 0.64 eV. The calculated width of the depletion region is $W= 684\text{ nm}$, and therefore the 880 nm Si membrane is not fully depleted. **Fig. R19** has been modified to better illustrate the band structure. Since W is less than $1\text{ }\mu\text{m}$ with the updated $q\phi_{Bn}$, the sentence about that “the space-charge region of a Schottky junction can reach several micrometers in width” has been deleted in the revised manuscript.

Fig. R19 Illustration of the basic operating principle of the transistor.

Corresponding modification has been made in Fig. 1 in the revised manuscript.

4/ I do not understand the statement on p. 12 that ohmic contacts were obtained due to the generation current at the etched surface. It is also not clear from the

supplementary material (Fig. 1). In Fig. 1c, one can indeed observe that after prolonged etching (7 min) ohmic behavior is found. But I do not agree with the explanation or at least I do not understand it. It is known that it is easier to make a good ohmic contact to a damaged surface, by defect-assisted leakage, so this could be the reason. At the same time, it is known that dry etching damage can lead to the introduction of shallow donors, so that the doping density can be changed. By the way, I presume that the silicon membrane is p-type (the SOI film?). I would suggest to verify the resistivity of the membrane if possible to check whether there are some changes in the net doping density. This will also have a drastic impact on the fully or partially depleted nature of the 880 nm film.

Response:

Thank you very much for pointing out the possible mechanism of Ohmic contact and valuable suggestions to investigate the doping concentration of Si. According to your comment and a reference¹⁴ (Schroder, D. K. *Semiconductor Material and Device Characterization, Third Edition* (John Wiley & Sons, Inc., New Jersey, 2006), page 129.), the mechanism of Ohmic contact has been changed as follows: sufficient RIE damages the Si surface where defect-assisted leakage and high recombination rates lead to the Ohmic contact.

Fig. R20 Resistivity measurements of the RIE-etched Si layer on a SOI wafer based on a transfer length method. **a** Optical micrograph of a Si strip with deposited Au electrodes in different distances of 8, 12, 17, 23 and 30 μm. **b** I-V characteristics between adjacent Au electrodes. **c** Relationship of the resistance and distance.

According to your suggestion, the resistivity of n-Si after RIE was investigated by a transfer length method test (**Fig. R20**)¹⁴. The I-V characteristics between adjacent Au electrodes indicate an Ohmic contact. The resistance and distance have a linear relationship as fitted (dash line in **Fig. 20c**). The slope equals to $\rho/(Wt)$, where ρ is the resistivity of Si, W is the width of Au electrode (93 μm), and t is the thickness of the Si trip (about 880 nm) (**Supplementary Fig. 15**). ρ was fitted to be 4.5 Ω·cm which is similar to the original SOI wafer (1-6 Ω·cm), indicating that the RIE processes does not change the doping concentration of Si obviously.

Corresponding modification has been made in Page 14 (Lines 8-11) in the revised manuscript as well as Supplementary Figs. 1 and 2 in the revised Supplementary Information.

Minor detail: p. 2 supplementary line 33: can now tunnel through the collector

barrier to be collected. This is a strange sentence - perhaps to be collected can be removed or replaced.

Response:

Thanks for your kind correction. The sentence has been changed to:

“as V_c increases, the current gain in **Fig. 3g** significantly increases since more emitted electrons can tunnel through the collector barrier.”

Corresponding modification has been made in Page 19 (Lines 11-13 in red color) in the revised Supplementary Information.

Reference

1. Di Lecce, V. et al. Graphene-base heterojunction transistor: an attractive device for terahertz operation. *IEEE Trans. Electron Dev.* **60**, 4263-4268 (2013).
2. Guo, H. et al. All-two-dimensional-material hot electron transistor. *IEEE Electron Dev. Lett.* **39**, 634-637 (2018).
3. Zubair, A. et al. Hot electron transistor with van der Waals base-collector heterojunction and high-performance GaN emitter. *Nano Lett.* **17**, 3089-3096 (2017).
4. Jin, Z., Li, X., Mullen, J. T. & Kim, K. W. Intrinsic transport properties of electrons and holes in monolayer transition-metal dichalcogenides. *Phys. Rev. B* **90**, 045422-1-7 (2014).
5. Tongay, S. et al. Rectification at graphene-semiconductor interfaces: zero-gap semiconductor based diodes, *Phys. Rev. X* **2**, 011002-1-10 (2012).

6. Sinha, D. & Lee, J. U. Ideal graphene/silicon Schottky junction diodes. *Nano Lett.* **14**, 4660-4664 (2014).
7. Parui, S. et al. Temperature dependent transport characteristics of graphene/n-Si diodes. *J. Appl. Phys.* **116**, 244505-1-5 (2014).
8. Miao, X.H. et al. High efficiency graphene solar cells by chemical doping, *Nano Lett.* **12**, 2745-2750 (2012).
9. Shi, E. et al. Colloidal antireflection coating improves graphene-silicon solar cells, *Nano Lett.* **13**, 1776-1781 (2013).
10. Kim, H.-Y., Lee, K., McEvoy, N., Yim, C. & Duesberg, G.S. Chemically modulated graphene diodes. *Nano Lett.* **13**, 2182-2188 (2013).
11. Singh, A., Uddin, M. A., Sudarshan, T. & Koley, G. Tunable reverse-biased graphene/silicon heterojunction Schottky diode sensor. *Small* **10**, 1555-1565 (2014).
12. Yang, H. et al. Graphene barristor, a triode device with a gate-controlled Schottky barrier. *Science* **336**, 1140-1143 (2012).
13. Di Bartolomeo, A. Graphene Schottky diodes: an experimental review of the rectifying graphene/semiconductor heterojunction. *Phys. Reports* **606**, 1-58 (2016).
14. Schroder, D. K. *Semiconductor Material and Device Characterization, Third Edition* (John Wiley & Sons, Inc., New Jersey, 2006).
15. Urteaga, M., Griffith, Z., Seo, M., Hacker, J., & Rodwell, M. J. W. InP HBT technologies for THz integrated circuits. *Proc. IEEE* **105**, 1051-1067 (2017).
16. Urteaga, M. et al. 130nm InP DHBTs with $f_t > 0.52\text{THz}$ and $f_{\text{max}} > 1.1\text{THz}$. *Proc.*

69th Annu. Device Res. Conf., 281-282 (2011).

17. Schröter, M. et al. Physical and electrical performance limits of high-speed SiGeC HBTs-part I: vertical scaling. *IEEE Trans. Electron Dev.* **58**, 3687- 3696 (2011).
18. Schröter, M. et al. Physical and electrical performance limits of high-speed SiGeC HBTs-part II: lateral scaling. *IEEE Trans. Electron Dev.* **58**, 3697-3706 (2011).
19. Sze, S. M. & Ng, K. K. *Physics of Semiconductor Devices, Third Edition* (John Wiley & Sons, Inc., New Jersey, 2007).
20. Atalla, M. M. & Soshea, R. W. Hot-carrier triodes with thin-film metal base. *Solid-State Electron.* **6**, 245-250 (1963).
21. Di Lecce, V. et al. Graphene-base heterojunction transistor: an explorative study on device potential, optimization, and base parasitics. *Solid-State Electron.* **114**, 23-29 (2015).
22. Kiefer, A. M. et al. Si/Ge junctions formed by nanomembrane bonding,” *ACS Nano* **5**, 1179-1189 (2011).
23. Zhang, Z. et al. Rosin-enabled ultraclean and damage-free transfer of graphene for large-area flexible organic light-emitting diodes. *Nat. Commun.* **8**, 14560-1-9 (2017).
24. Wang, X. et al. Direct observation of poly(methyl methacrylate) removal from a graphene surface. *Chem. Mater.* **29**, 2033–2039 (2017).
25. Mehr, W. et al. Vertical graphene base transistor. *IEEE Electron Dev. Lett.* **33**, 691-693 (2012).

Reviewers' comments:

Reviewer #1 (Remarks to the Author):

The authors addressed some of the concerns in their reply with additional experimental data and explanations. I appreciate that they fabricated devices with metal contacts, provided sheet resistance measurement, cited appropriate literature and removed the claim for record current density. The additional information provided by the authors help us to construct a better picture about the novelty, impact and relevance to the broader readership.

In this manuscript, the authors successfully demonstrated working transistors and claimed orders of magnitude potential performance improvement by extrapolation. However, they didn't provide enough supporting experimental evidence to support such optimistic claims. Although the current work can be considered first demonstration of working graphene-base transistor with Schottky emitter but it requires further device engineering and characterization to claim significant advancement over the state-of-the-art. Unfortunately, I can't recommend the manuscript for Nature Communications in the current version. My comments are given below,

1. The key contribution of this manuscript is the experimental demonstration of graphene-base heterojunction transistor theoretically proposed by Di Lecce et al. (IEEE TRANSACTIONS ON ELECTRON DEVICES, VOL. 60, NO. 12, DECEMBER 2013). The theoretical work predicted that the use of Schottky junction emitter instead of tunnel emitter would improve the high frequency performance by both improving the current density and capacitance. The results presented in the manuscript shows maximum emitter current density of 692 A/cm^2 at $V_E = -5\text{V}$ which is higher than best reported ($\sim 230 \text{ A/cm}^2$) graphene-base transistor with tunnel emitter. However, the authors claim in abstract and later in the manuscript that "Such Schottky emitter shows a current of 692 A/cm^2 and a capacitance of 28 nF/cm^2 " which is ambiguous. The authors theoretically estimated the capacitance to be 28 nF/cm^2 and they didn't report the experimental value in the manuscript. The authors should report the experimental value to claim that their emitter shows certain value of capacitance.
2. The estimation of cutoff frequency above 1 GHz is oversimplified as the authors ignore the contribution of base-collector capacitance and significantly large series resistance present in the reported device. The authors need to report the experimental value of base-collector capacitance and series resistance to provide a realistic estimation of cutoff frequency of the presented device.
3. Could the authors please discuss the differences between Gr/n+ Ge diode presented in the current version (Fig. 3e) and previous version (Fig. S3b) of the manuscript? Did the authors change the fabrication process significantly?
4. The use of g_m as the emitter conductance can be confusing to the readers. In standard literature, the transconductance of a transistor is represented by g_m . I would recommend to avoid the use of g_m as emitter conductance.

Reviewer #2 (Remarks to the Author):

Let me thank the Authors for their careful response, with which I am satisfied. In my opinion, the paper can now be published. I would suggest only some minor optional adjustments, which do not require further review:

1. The Schottky barrier heights have been extracted at non-zero voltage. Due to the image force, the barriers are voltage dependent. The Authors may have misunderstood my remark in the previous report, because I referred there to the expression for the simple case of emission to vacuum and did not comment on this methodological oversimplification. But if the barrier heights are extracted for a few voltages, just as reported in the manuscript for -0.1 eV, and then are plotted against the voltage, a clear-cut tendency is seen and extrapolation to $V=0$ becomes straightforward. Taking the data from Supplementary Fig. 3 for several voltages in the range from 0.025 to 0.25 V, I estimated the barrier at $V=0$ to be 0.68 eV (see the attached plot). This was done with linear regression; the correct dependence is of the type $\sqrt{E_{\text{max}}}$, where E_{max} is the maximum electric field at the interface, and it would slightly increase the estimated value. The height of 0.68 eV is very close to 0.64 eV quoted by the Authors, but the whole procedure yields also a feeling about the accuracy (please note, how noisy the IV curves are). The Authors may want to consider this in their final manuscript.

2. The paper reads nicely, but I think it would profit from language corrections. I am not a native English speaker, but for example there are sentences in which I surely would correct the articles.

Reviewer #3 (Remarks to the Author):

The Authors have done a serious effort in replying to the Reviewers' comments and improved their manuscript significantly. I have no further comments.

Responses to the comments of reviewers

Reviewer #1:

The authors addressed some of the concerns in their reply with additional experimental data and explanations. I appreciate that they fabricated devices with metal contacts, provided sheet resistance measurement, cited appropriate literature and removed the claim for record current density. The additional information provided by the authors help us to construct a better picture about the novelty, impact and relevance to the broader readership. In this manuscript, the authors successfully demonstrated working transistors and claimed orders of magnitude potential performance improvement by extrapolation. However, they didn't provide enough supporting experimental evidence to support such optimistic claims. Although the current work can be considered first demonstration of working graphene-base transistor with Schottky emitter but it requires further device engineering and characterization to claim significant advancement over the state-of-the-art. Unfortunately, I can't recommend the manuscript for Nature Communications in the current version. My comments are given below,

Response:

Thank you very much for your review. In accordance with the reviewer's valuable comments, we have addressed all the concerns of the reviewer in the revised manuscript.

Details are discussed as follows.

1. The key contribution of this manuscript is the experimental demonstration of

graphene-base heterojunction transistor theoretically proposed by Di Lecce et al. (IEEE TRANSACTIONS ON ELECTRON DEVICES, VOL. 60, NO. 12, DECEMBER 2013). The theoretical work predicted that the use of Schottky junction emitter instead of tunnel emitter would improve the high frequency performance by both improving the current density and capacitance. The results presented in the manuscript shows maximum emitter current density of 692 A/cm^2 at $V_E = -5\text{V}$ which is higher than best reported ($\sim 230 \text{ A/cm}^2$) graphene-base transistor with tunnel emitter. However, the authors claim in abstract and later in the manuscript that “Such Schottky emitter shows a current of 692 A/cm^2 and a capacitance of 28 nF/cm^2 ” which is ambiguous. The authors theoretically estimated the capacitance to be 28 nF/cm^2 and they didn’t report the experimental value in the manuscript. The authors should report the experimental value to claim that their emitter shows certain value of capacitance.

Response:

Thank you for the valuable comment. As the reviewer pointed out, the conclusion will be better if supported by an experimental value of the emitter capacitance. According to this comment, the emitter capacitance has been measured by a semiconductor analyzer (Agilent B1500A with a capacitance measurement unit B1500A-A20). The C-V curve of the Si-Gr at a forward bias (Gr is connected to the ground) shows that the peak capacitance is about 41 nF/cm^2 at -0.30 V (measured at a frequency of 100 kHz) (Fig. R1a), which is consistent with the theoretical estimation (Supplementary Fig. 4). Statement of the capacitance value and Fig. 2d has been updated with 41 nF/cm^2 . Multiple emitters have been measured and the peak

capacitance is between 30 and 50 nF/cm². At the forward bias of -0.30 V, the capacitance-frequency characteristics (Fig. R1b) shows that the capacitance is stable up to 100 kHz, and the measurement result is affected by interface and series resistance at frequency beyond 100 kHz¹. Thus, the frequency used to measure the capacitance is selected to be 100 kHz.

Fig. R1 Capacitance measurement of a Si-Gr Emitter. **a** The C-V characteristic of the Si-Gr emitter obtained from experiments at a frequency of 100 kHz. **b** The C-f characteristic of the Si-Gr emitter at a forward bias of -0.30 V.

Corresponding modification has been made in Page 1 (Line 18), Page 3 (Line 14), Page 6 (Line 22), Page 13 (Line 13), Page 16 (Line 14) and Fig. 2d in the revised manuscript as well as Supplementary Fig. 4 in the revised Supplementary Information in red color.

2. The estimation of cutoff frequency above 1 GHz is oversimplified as the authors ignore the contribution of base-collector capacitance and significantly large series resistance present in the reported device. The authors need to report the experimental value of base-collector capacitance and series resistance to provide a realistic estimation of cutoff frequency of the presented device.

Response:

Thank you very much for the valuable comment. As the reviewer pointed out, an

estimation of the alpha cut-off frequency f_α with more details and experimental evidence will further benefit the conclusion in the paper. According to your suggestion, f_α has been analyzed and estimated considering the experimental result of emitter capacitance, collector capacitance and series resistance as below.

Firstly, it should be emphasized that the f_α estimated here is the intrinsic f_α . Only delay time contributed by junction capacitance will be included, and the effect of the parasitic electrode capacitance should be excluded as discussed in a reference². In our device, the large area Au and graphene on top of the 30-nm-thick Al₂O₃-Ge substrate (Gr/Au-Al₂O₃-Ge) (Fig. 1) will induce large parasitic capacitance which is orders-of-magnitude larger than the Gr-Ge collector junction capacitance. It is not useful for implementing device functions, and in a device for production, it can be reduced by for example, mesa Ge structure together with thick insulating layer and small base electrode. In the following estimation of f_α , this parasitic capacitance is excluded.

Estimation of τ_e

The intrinsic alpha cut-off frequency f_α can be expressed as $f_\alpha=1/[2\pi(\tau_e+\tau_b+\tau_c)]$. The base transit time τ_b is ignored ($\tau_b=0$) thanks to the atomically thin thickness of graphene. The emitter charging time τ_e is calculated as $\tau_e=C_e/g_e$ where $C_e=41$ nF/cm² is the peak emitter capacitance (Fig. R1a) and g_e is the emitter conductance which can be achieved from the I-V characteristics of the emitter junction (347 S/cm² at -5 V) (Fig. 2a), leading to a $\tau_e=C_e/g_e=118$ ps at -5 V.

Estimation of τ_{cc}

The collector delay time τ_c can be expressed as $\tau_c=\tau_{ct}+\tau_{cc}$ where τ_{ct} is the collector

(depleted region) transit time and τ_{cc} is the collector charging time. We will first estimate τ_{cc} with experimental results. τ_{cc} is estimated by $\tau_{cc}=r_c C_c$ where C_c is the collector capacitance and r_c is the collector series resistance. As discussed, to estimate the intrinsic f_α , the effect of the parasitic electrode capacitance should be excluded, thus C_c cannot be directly measured from the Gr-Ge junction in the device, because the result will contain the parasitic electrode capacitance. At the same time, r_c cannot be directly measured from the I-V characteristics of the Gr-Ge junction either, because the measured series resistance of the junction will contain not only the collector series resistance, but also the base series resistance.

To determine C_c and r_c , we have fabricated Au(50nm)/Ti(5nm)-Ge Schottky junctions (Fig. R2a) using the same n^+ -Ge substrate as the device in Fig. 3e, where the area of Au/Ti is the same as the Gr- n^+ -Ge junction area, i.e. the area of the window of Al_2O_3 (Fig. 1b). We measure the series resistance (Fig. R2b) and the capacitance at a reverse bias of 4 V (Fig. R2c,d) of the Au/Ti-Ge junctions, and these results will be r_c (about 5 Ω) and C_c (1.7 pF), leading to a $\tau_{cc}=r_c C_c=8.5$ ps. The reason is discussed below.

Fig. R2 Series resistance and capacitance measurement of a Au/Ti-Ge Schottky junction. **a** I-V characteristic of the junction. The Au/Ti is connected to the ground. Inset: an optical image of the junction (scale bar: 20 μm). **b** Differential resistance r calculated from the I-V characteristic at a forward bias, which tends to the series resistance about 5 Ω . **c** The C-V characteristic of the junction at reverse bias at a frequency f of 100 kHz. As the reverse bias increases, C decreases because the depleted region width increases. **d** The C-f characteristic of the junction at a reverse bias of 4 V. The capacitance is stable up to 100 kHz. 100 kHz is selected.

1. The series resistance of the Au/Ti-Ge Schottky junction is the sum of a) the resistance integrated over the quasi-neutral region (between the depletion-layer edge and bottom Au electrode), b) the spreading resistance in the substrate, and c) the resistance due to the bottom Au ohmic contact with the substrate, if the resistance of Au(50nm)/Ti(5 nm) is ignored³. For the Gr-Ge junction, the collector resistance r_c is also the sum of the above 3 items. Since the same Ge substrate with bottom Au electrode is used for the Au-Ge junction and the Gr-Ge junction and the junction areas are also the same, the latter two items (b and c) should be the same for Gr-Ge junction and Au-Ge junction. The width of the depletion-layer is different for the two junctions, but only

with a difference less than 1 μm . The thickness of the Ge substrate is about 500 μm , thus the distance between the depletion-layer edge and bottom Au electrode is almost the same, and the first item (a) should also be the same. Based on the above analysis, the measured series resistance of the Au/Ti-Ge Schottky junction is the same as the Ge series resistance in the Gr-Ge junction r_c .

2. At a reverse bias V , the capacitance of a Schottky junction can be calculated³ as $C = \epsilon_{\text{Ge}} \epsilon_0 / W$, $W = (2 \epsilon_{\text{Ge}} \epsilon_0 / q N_d (\psi_{\text{bi}} + V - V_t))^{0.5}$, $\psi_{\text{bi}} = \phi_{\text{Bn}} - V_n$, $V_n = V_t \cdot \ln(N_c / N_d)$, where ϵ_{Ge} is the relative dielectric constant of Ge, ϵ_0 is the permittivity in vacuum, W is the width of the depletion region, q is elementary charge, N_d is the doping concentration of Ge, ψ_{bi} is the potential barrier height at the Ge side when no bias is applied, V_t is the thermal voltage, $q \phi_{\text{Bn}}$ is the Schottky barrier height, V_n is the Fermi potential of Ge, and N_c is the effective density of states in conduction band. The same Ge substrate is used for Au-Ge and Gr-Ge junctions, thus the difference of the capacitance comes from ϕ_{Bn} which is 0.52 V for Ti-Ge⁴ and 0.22 V for Gr-Ge (Supplementary Fig. 7). However, at a large reverse bias, for example $V=4$ V, the difference of ϕ_{Bn} (0.30 V) can be ignored, thus the difference of W can be ignored. The measured capacitance of Au/Ti-Ge junction is the intrinsic capacitance C_c of the Gr-Ge collector junction.

Estimation of τ_{ct}

Next, τ_{ct} is estimated³ by $\tau_{\text{ct}} = \chi / (2\nu)$ where χ is the width of the depletion region of the collector junction and ν is saturation velocity in Ge (7×10^6 cm/s). χ is estimated by $\chi = \epsilon_{\text{Ge}} \epsilon_0 / C_c = 3.0 \times 10^{-5}$ cm where C_c is the measured collector capacitance (1.7 pF or say 4.7×10^{-8} F/cm² at 4 V in Fig. R2c), leading to a $\tau_{\text{ct}} = \chi / (2\nu) = 2.1$ ps.

Estimation of f_α

Based on the above experimental results and analysis, the intrinsic alpha cut-off frequency f_α has been updated in Fig. R3 at a bias of $V_c=4$ V by 1) using the experimental result of the emitter capacitance $C_e=41$ nF/cm² to replace the previous theoretical value, 2) adding the collector charging time $\tau_{cc}=8.5$ ps, and 3) adding the collector transit time $\tau_{ct}=2.1$ ps. With a bias of $V_e=-5$ V and $V_c=4$ V, $f_\alpha=1/[2\pi(\tau_e+\tau_b+\tau_{cc}+\tau_{ct})]=1/[2\pi(118+0+8.5+2.1)$ ps]=1.2 GHz. For the transistors with tunnel emitters, only emitter charging time is used to estimate their f_α , which may decrease when collector delay time is involved.

Fig. R3 Comparison of f_α of graphene-base transistors with different emitters. The one with the Si-Gr Schottky emitter shows the best cut-off frequency of 1.2 GHz. (The reference numbers in this figure is for the main text.)

Corresponding modification has been made in Page 7 (Lines 3, 5-6, 9), Page 13 (Line 12) in red color and Fig. 2d in the revised manuscript as well as Supplementary Fig. 5 in the revised Supplementary Information.

3. Could the authors please discuss the differences between Gr/n⁺ Ge diode presented in the current version (Fig. 3e) and previous version (Fig. S3b) of the manuscript? Did the authors change the fabrication process significantly?

Response:

Thank you for your comments. We are sorry for the confusion. The following 2 points should be stressed. First, the n⁺-Ge substrate has been changed to improve the performance: in the original version (Fig. S3b) the resistivity is ~0.01 Ωcm while in the current version (Fig. 3e) the resistivity is ~0.1 Ωcm. Second, in the original version (Fig. S3), the unit for the current should be “A/cm²”, not “A”.

4. The use of g_m as the emitter conductance can be confusing to the readers. In standard literature, the transconductance of a transistor is represented by g_m . I would recommend to avoid the use of g_m as emitter conductance.

Response:

Thank you for the comments. According to this kindly suggestion, the emitter conductance is represented by g_e now.

We have changed all “ g_m ” to “ g_e ” in red color in the revised manuscript and Supplementary Information.

Reviewer #2: Let me thank the Authors for their careful response, with which I am satisfied. In my opinion, the paper can now be published. I would suggest only some minor optional adjustments, which do not require further review:

Response:

Thank you very much for your review.

1. The Schottky barrier heights have been extracted at non-zero voltage. Due to the image force, the barriers are voltage dependent. The Authors may have misunderstood my remark in the previous report, because I referred there to the

expression for the simple case of emission to vacuum and did not comment on this methodological oversimplification. But if the barrier heights are extracted for a few voltages, just as reported in the manuscript for -0.1 eV, and then are plotted against the voltage, a clear-cut tendency is seen and extrapolation to $V=0$ becomes straightforward. Taking the data from Supplementary Fig. 3 for several voltages in the range from 0.025 to 0.25 V, I estimated the barrier at $V=0$ to be 0.68 eV (see the attached plot). This was done with linear regression; the correct dependence is of the type $\sqrt{E_{\max}}$, where E_{\max} is the maximum electric field at the interface, and it would slightly increase the estimated value. The height of 0.68 eV is very close to 0.64 eV quoted by the Authors, but the whole procedure yields also a feeling about the accuracy (please note, how noisy the IV curves are). The Authors may want to consider this in their final manuscript.

Response:

Thank you very much for your valuable comments. Because of the image-force lowering and recombination at the interface, the equivalent potential barrier height is voltage dependent. According to this comment, using the recommended extrapolation method, the barriers of the Si-Gr junction at $V=0$ are determined as 0.68 eV (Fig. R4). As the reviewer pointed out, although this value is very close to the previous estimated values at -0.1 V, better accuracy has been achieved. The barrier-height-related values and figures have been updated in the revised manuscript, and each conclusion remains unchanged.

Fig. R4 Schottky barrier height estimation at $V=0$ by an extrapolation method.

Corresponding modification has been made in Page 6 (Lines 6-8), Page 8 (Line 15) in red color in the revised manuscript as well as Supplementary Fig. 3, 4, 7, 15 and Table 1 in the revised Supplementary Information.

2. The paper reads nicely, but I think it would profit from language corrections. I am not a native English speaker, but for example there are sentences in which I surely would correct the articles.

Response:

According to the reviewer's suggestion, we have polished English language usage in the revised manuscript.

Reviewer #3:

The Authors have done a serious effort in replying to the Reviewers' comments and improved their manuscript significantly. I have no further comments.

Response:

Thank you very much for your review.

Reference

1. Şahin, B., Çetin, H. & Ayyildiz, E. The effect of series resistance on capacitance–voltage characteristics of Schottky barrier diodes. *Solid State Commun.* **135**, 490–495 (2005).
2. Liao, Lei et al. High-speed graphene transistors with a self-aligned nanowire gate. *Nature* **467**, 305–308 (2010).
3. Sze, S. M. & Ng, K. K. *Physics of Semiconductor Devices, Third Edition* (John Wiley & Sons, Inc., New Jersey, 2007).
4. Nishimura, T., Kita, K. & Toriumi, A. Evidence for strong Fermi-level pinning due to metal-induced gap states at metal/germanium interface. *Appl. Phys. Lett.* **91**, 123123-1-3 (2007).

REVIEWERS' COMMENTS:

Reviewer #1 (Remarks to the Author):

I would like to thank the authors for new experiments to address my concerns. I have no further question.